

# MonARCh: an actor based architecture for dynamic linked data monitoring

Burak Yönyül, Oylum Alatlı and Rıza Cenk Erdur

Faculty of Engineering, Department of Computer Engineering, Ege University, Izmir, Turkey

## ABSTRACT

Monitoring the data sources for possible changes is an important consumption requirement for applications running in interaction with the Web of Data. In this article, MonARCh which is an architecture for monitoring the result changes of registered SPARQL queries in the Linked Data environment, is proposed. MonARCh can be comprehended as a publish/subscribe system in the general sense. However, it differs in how communication with the data sources is realized. Data sources in the Linked Data environment do not publish the changes in the data. MonARCh provides the necessary communication infrastructure between the data sources and the consumers for the notification of changes. Users subscribe SPARQL queries to the system which are then converted to federated queries. MonARCh periodically checks for updates by re-executing SERVICE clauses and notifying users in case of any result change. In addition, to provide scalability, MonARCh takes the advantage of concurrent computation of the actor model. The parallel join algorithm utilized speeds up query execution and result generation processes. The design science methodology is used during the design, implementation and evaluation of the architecture. When compared to the literature MonARCh meets all the sufficient requirements from the linked data monitoring and state of the art perspectives while having many outstanding features from both points of view. The evaluation results show that even while working under the limited two-node cluster setting MonARCh could reach from 300 to 25,000 query monitoring capacity according to the diverse query selectivities executed within our test bench.

## INTRODUCTION

The Linked Data concept was introduced by Tim Berners-Lee in his technical note (*Berners-Lee, 2006*). In this technical note, he manifested the basic principles for making the data published on the Web a part of the global information space, which is usually referred to as the "Web of Data" (*Bizer, Heath & Berners-Lee, 2011*). Since then, the Web of Data has gradually expanded as organizations from different domains have begun to publish their data following the Linked Data principles. Today, the Web of Data contains a large number of interrelated data sets covering specific domains such as government, media, entertainment, life sciences and geographical places. In addition to them, there are data sets that span over multiple domains such as DBpedia (https://www.dbpedia.org)

Corresponding author
Burak Yönyül,
burak.yonyul@ege.edu.tr

(*Lehmann et al., 2015*). All of these datasets are referred to as Linked Open Data (LOD) Cloud (https://lod-cloud.net).

As the Web of Data got bigger, a need for querying emerged. This need brought the introduction of SPARQL. SPARQL, which is the standard query language for Linked Data today, became an official W3C Recommendation on 15 January 2008 (*Prud'hommeaux & Seaborne, 2017*). SPARQL 1.0 allowed querying individual data sets by means of SPARQL endpoints. However, it had no means of running federated queries that could combine data coming from several SPARQL endpoints at once. Today, the ability to process federated queries is an obvious requirement for consuming Linked Data distributed over the Web. Federated querying capability was added to SPARQL in version 1.1 (*Arenas & Pérez, 2012*).

Although the ability to execute federated queries addressed an important need for Linked Data consuming applications, querying alone was not enough for providing all of the data these applications needed. As the number of organizations serving Linked Open Data increased, it became a critical task for applications to monitor changes in the data to be able to react to these changes (for example, a list of links to recent newspaper articles about companies whose stock market value has increased by ten percent in the past day) on time. Thus, communication of data dynamics became an important issue (*Umbrich, Villazón-Terrazas & Hausenblas, 2010*) for the continuously expanding and dynamically changing Web of Data (*Käfer et al., 2013*; *Käfer, 2019*).

In the work by *Umbrich, Villazón-Terrazas & Hausenblas (2010)*, four use cases which require managing and communicating the dynamics of the Linked Data on the Web are discussed. Two of these are synchronisation and smart caching. Here, synchronisation refers to change notification and smart caching refers to keeping the necessary data for an application in a local cache. This work further decomposes each use case and lists the requirements for each. These requirements are grouped into five as change description method, granularity level of the change, communication mechanism for change notification (the publish/subscribe mechanism is recommended), change discovery mechanism and scalability. The suggested granularity levels are dataset, resource, statement and graph structural levels. Among the RDF (https://www.w3.org/RDF) change notification systems examined in this study, only WebSub Protocol (*Genestoux & Parecki, 2018*) supports a push-pull based approach. None of them, however, support change notification for federated SPARQL queries.

This requirement is also addressed in a comprehensive survey by *Klímek, Škoda & Nečaský (2019)*, where the requirements for a Linked Data Consumption Platform (LDCP) have been identified. In this work, an LDCP is defined as a platform to be used by users who are not Linked Data experts. However, tools that can be combined with an LDCP to extend its linked Data consumption features are included too. That is why, tools that provide machine readable RDF output are taken into consideration as well. In the survey, the authors define this requirement as the ability to periodically monitor a data source and trigger actions when the data source changes (please see requirement 15 in *Klímek, Škoda & Nečaský (2019)*). This survey also introduces and evaluates sixteen tools

(*Klímek, Škoda & Nečaskỳ, 2019*). But none of them addresses the Linked Data monitoring requirement (please see Tables 3 and 4 in *Klímek, Škoda & Nečaskỳ (2019)*).

In the literature, there are some push-pull systems that attempt to address this issue, which will be discussed in "Related Work". However, they either target single SPARQL endpoints (*Passant & Mendes, 2010*; *Popitsch & Haslhofer, 2011*; *Roffia et al., 2018*) or are not designed for propagating RDF updates to client applications (*Barazzutti et al., 2013*, *2014*; *Dossot, 2014*; *Li et al., 2011*; *Ma et al., 2014*; *Ma, Wang & Pei, 2014*; *Rowstron et al., 2001*; *Setty et al., 2012*; *Snyder, Bosnanac & Davies, 2011*). Applications in today's highly distributed Web of Data need a scalable service for isolating the developers and the applications they develop from this monitoring burden. Such a service should have an infrastructure that can pull and join the data they need from diverse SPARQL endpoints, monitor the results for changes and push them as needed to its clients. To our knowledge, such a system is not present in the literature yet. MonARCh (Monitoring Architecture for sparql Result Changes) is the first service that addresses Linked Data monitoring requirements at the organisational level.

MonARCh, as previously stated, is a monitoring architecture for federated SPARQL queries in the Linked Data environment. Because monitoring requests can come from a large number of asynchronous applications, scalability is a critical consideration when designing such an architecture. The actor model is a well-designed solution for concurrent computation problems that require scalability. MonARCh is therefore built on top of the actor model (*Hewitt, Bishop & Steiger, 1973*; *Hewitt, 2010*).

Other considerations in developing such an architecture include the basic entities to be monitored and the underlying communication mechanism of the monitoring architecture. In MonARCh, the aim is to provide just the data necessary for a client application. There may be multiple changes in the datasets involved in a SPARQL query. However, registered applications often need to know the combined information synthesized from the data coming from different SPARQL endpoints. The developer still has the responsibility of removing and merging the relevant data if the client applications are informed of every modification in the data originating from individual SPARQL endpoints. Therefore, the fundamental entities that are tracked and combined in MonARCh are not entire datasets but rather the results of particular SPARQL (*Consortium, 2013*) queries.

Monitoring infrastructures are usually based on either the pull or the push approach. In the pull approach, the consumer monitors the datasets by means of periodic queries. In the push mechanism, the consumer subscribes to the data publisher and then the data publisher acknowledges the changes to subscribers. MonARCh combines both approaches and uses a combined push-pull mechanism. While the pull operation defines the querying the Web of Data for updates, the push operation stands for the notification of SPARQL result changes to the registered clients. Using the pull approach, all new datasets are dynamically discovered each time before querying. Applications, which need to monitor some specific part of the Web of Data, register their requests through SPARQL queries to the monitoring service. They are notified in case of a resultset change. Using this combined push-pull approach constitutes one of the novel sides of MonARCh.

The other novel side of the proposed monitoring architecture is the ability to separate a complex query into its SERVICE clauses and execute each one of them independently. These SERVICE clauses are executed based on the change frequencies of their related data sets. For this purpose, it makes use of DaDy metadata (http://purl.org/NET/dady#) (*Hausenblas, 2010*). DaDy metadata resides in VoID (*Alexander et al., 2009*) documents of Linked Data sources. It defines the change frequencies of those sources. The partial results may come either from a newly executed SERVICE clause, or from the local partial result caches of previously executed SERVICE clauses. These partial results are then merged using a parallel hash join algorithm. Thus, monitoring each SERVICE clause separately allows the system to reuse the cached partial result of a particular SERVICE clause. These cached resultsets can be combined to obtain the final result of complex queries that contain them.

To summarize, the prominent and distinguishing features of MonARCh are actor-based scalability, a combined push-pull approach in the underlying architecture, and the ability to monitor SERVICE clauses independently based on dataset change frequencies.

MonARCh is developed and evaluated following the design science methodology defined in *Wieringa (2014)*. A case study that uses DBpedia, New York Times and stock market datasets is developed following this methodology. Each dataset has a different change frequency which allows to test whether MonARCh can monitor each simple SPARQL query that constitutes the federated query at different intervals, cache the results of each, join the results effectively and notify the client applications on time.

The rest of the article is organized as follows. "Background" defines the requirements of the Linked Data monitoring environment, which constitute this article's core idea. "Related Work" studies the related work present in the literature. "Requirements of the Linked Data Monitoring Architecture" defines the functional and non functional requirements of MonARCh in detail. "Design of the Abstract Architecture and the Validation Model" discusses the abstract architecture design. "Implementation of the Validation Model" and "Evaluation and Validation of the Architecture" elaborate on the implementation and evaluation details of MonARCh respectively. Lastly, "Conclusion and Future Work" summarizes the whole work and proposes some possible improvements.

## BACKGROUND

### Linked data relevant background

The foundations of this study lay in monitoring changes in the SPARQL query results in the linked data environment. Then it would be rational to clarify the relevant concepts to understand the underlying theory and the motivation better. Linked Data is a paradigm that emphasizes the publication and interlinking of structured data on the web in a standardized format, typically using technologies like RDF (Resource Description Framework) and URIs (Uniform Resource Identifiers). This approach enables data to be connected and queried across different sources and domains, forming a global network of interrelated information. Linked Data adheres to a set of principles known as the "Linked Data Principles," which include using HTTP URIs to identify resources, making resource descriptions available at these URIs using standard formats such as RDF, and including

links to other resources to enable navigation and discovery of related information. This facilitates data integration, interoperability, and knowledge discovery across heterogeneous data sources.

SPARQL is a query language used to retrieve and manipulate data stored in RDF (Resource Description Framework) format. RDF is a standardized way of representing data as triples, consisting of subject-predicate-object statements in graph structure. SPARQL provides a powerful and expressive syntax for querying RDF data. It allows users to specify patterns of triples they want to match in the data graph, along with conditions and constraints. Queries are written in a declarative manner, where users describe what they want to retrieve rather than specifying how to retrieve it.

A federated SPARQL query is a type of SPARQL query that allows data to be queried across multiple distributed RDF data sources. Unlike a traditional SPARQL query, which operates on a single RDF dataset, a federated SPARQL query can access and integrate data from multiple RDF datasets located on different servers or endpoints. In a federated SPARQL query, the query is split into service clauses, each of which is executed on a separate RDF data source. The results from these service clauses are then combined or merged to produce the final result set. This enables querying and analyzing data that is distributed across different domains, organizations, or systems. Federated SPARQL queries use the SERVICE keyword to specify remote endpoints or data sources where service clauses should be executed. These endpoints can be SPARQL endpoints exposing RDF data or other types of data sources that support SPARQL query execution. Federated SPARQL queries are particularly useful in scenarios where data is distributed across multiple sources and needs to be integrated or analyzed collectively. They enable federated data querying and integration across disparate data sources, facilitating data interoperability, knowledge discovery, and cross-domain analysis.

Most of the data services which are available on the Web are commonly built on standards like REST (Representational State Transfer) and JSON (JavaScript Object Notation). Therefore monitoring infrastructures are heavily based on these web services. But there is still a gap among the linked data systems from the monitoring point of view. Therefore in this work MonARCh fills this gap by covering both the functional requirements about the monitoring task, and non functional requirements such as high performance and scalability.

## Design science relevant background

According to *Wieringa (2014)*, design science is the design and research of artifacts in context to improve that context. The methodology defined for this is referred to as Design Science Methodology (*Wieringa, 2014*), and consists of four main steps:

- Defining the design problem and knowledge questions to be answered by the research.
- Design cycle which consists of problem investigation, designing an artifact and defining the context it will operate in (artifact and its context are defined as a treatment), and the validation of the treatment defined.

- Defining the conceptual framework the research is based on and the theoretical generalizations that can be concluded at the end of the research.
- Empirical cycle aiming to test the treatment with empirical methods with the following steps: 1. problem analysis, 2. research setup design, 3. inference design, 4. validation of inferences against research setup, 5. research design, 6. research execution, 7. data analysis

Four empirical research methods can be utilized by the empirical cycle one of which is single case mechanism experiments (SCME). SCME are conducted with an object of study (OoS), which is a prototype model of the designed artifact placed in a model of the real-world context it will operate in. SCME are done in the lab so as to allow the researcher to have access to the architecture of OoS, explain its behaviour in terms of the architecture and validate the artifact design. These explanations are expected to be generalized to the population of all possible implementations of the artifact design and they will be validated by analogical reasoning. Since the research examined in this article aims to validate the abstract architecture designed for monitoring linked data changes, the SCME method is the most suitable one for this study.

## RELATED WORK

*Deolasee et al. (2001)* and *Bhide et al. (2002)* lay the foundation of the adaptive push-pull model. This model dynamically chooses either the push or the pull approach depending on the changing conditions. More recently, *Huang & Wang (2010)* used a model named combined push-pull in their study for resource monitoring in a cloud computing environment. The push model comes with a solution named the publish/subscribe architecture (*Eugster et al., 2003*). Publishers propagate their new data and subscribers register for some topics or content. A middleware layer seeks a match between the data sent by the publisher and the topics/contents registered by a subscriber. Later, it disseminates the relevant data from the publisher to the subscriber.

We want to separate the requirements for linked data and publish/subscribe domains, as well as analyze the available systems from various perspectives. Thus, we examine the related work in two different categories: the linked data relevant literature and the publish/subscribe relevant literature.

### Linked data relevant literature

The intended context of the system discussed in this article is the Semantic Web. There is neither restriction nor control over the changes in the Linked Data cloud. Additionally, the data sources are not obliged to inform their users about these changes. That's why the consumers do not have exact knowledge about the updates of the data sources. Therefore, firstly a linked data monitoring system should have the ability to work on the Web of Data. Moreover, because the data sources may probably be distributed over the Web, it should also have the ability to monitor federated SPARQL queries. Furthermore, the work involved in query generation would be simplified by the ability to convert raw requests into federated form. Because the frequency of the number of query monitoring requests can be

**Table 1 Coverage of the linked data specific requirements for the monitoring architecture by the literature in this context.**

| Platform | SPARQL | | | | | Working environment | | Monitoring as a service | Data acquisition | | Scalability |
|---|---|---|---|---|---|---|---|---|---|---|---|
| | Query monitoring | | | Result change | | | | | | | |
| | Single | Federated | Raw | Detection | Notification | Local | Web of data | | Publisher | Polling | |
| SPARQLES | | | | | | | ✓ | ✓ | | ✓ | |
| SparqlPUSH | ✓ | | | ✓ | ✓ | ✓ | | ✓ | ✓ | | |
| LDN | | | | ✓ | | | ✓ | | ✓ | ✓ | |
| DSnotify | ✓ | | | ✓ | ✓ | ✓ | | ✓ | ✓ | ✓ | |
| SEPA | ✓ | | | ✓ | ✓ | ✓ | | ✓ | ✓ | | |
| SPS | ✓ | | | ✓ | ✓ | ✓ | | ✓ | ✓ | | ✓ |
| MonARCh | ✓ | ✓ | ✓ | ✓ | ✓ | ✓ | ✓ | ✓ | | ✓ | ✓ |

dynamic and may increase over time, it is critical that such a system be scalable in order to facilitate adapting to the Web of Data's dynamic nature. Table 1 summarizes the literature's coverage of linked data-specific requirements.

In the literature, there are systems such as SPARQLES (*Vandenbussche et al., 2017*) which monitor SPARQL endpoints periodically. However, the main aim of these systems is to evaluate the availability, discoverability, performance and interoperability of the endpoints they monitor. While it can work as a service and operate on the Web of Data it has neither monitoring nor scalability requirements listed in the "Requirements of the Linked Data Monitoring Architecture" (see Table 1).

SparqlPuSH (*Passant & Mendes, 2010*) is a partial (push side) implementation of the WebSub (*Genestoux & Parecki, 2018*) protocol for the RDF data stores. WebSub (previously known as PubSubHubbub (*Fitzpatrick et al., 2013*)) is a web based publish/subscribe protocol that supports both pull and push based approaches. SPARQL queries are registered as a feed to a hub in SparqlPuSH, and the hub notifies subscribers when the results change. On the other hand, DSNotify (*Popitsch & Haslhofer, 2011*) is another publish/subscribe system which can either be used in a pull based or a push based manner, but not in a combined style. DSNotify can monitor both RDF graph areas and SPARQL query results. According to the preference, changes are either sent to the subscribers in a push based manner, or the users can query the changes in a pull based manner. Another study, SEPA (*Roffia et al., 2018*), is a SPARQL event processing architecture that detects and disseminates changes across the Web of Data using a content-based publish/subscribe mechanism. The study suggests the SPARQL 1.1 Update Language, Secure Event Protocol, and Subscribe Language, which are supported by the system's publishers and subscribers. SEPA enables the development of scalable systems with a design pattern by use of its brokering mechanism. On the other hand, SparqlPush uses the scalable Google public PuSH hub as a service, but does not have a PuSH hub as part of the system implementation. However, there is no concrete evidence about the scalability of these systems neither in their system architecture nor in their evaluation results. Therefore, they does not completely meet the "monitoring federated SPARQL queries" and "scalability"

**Table 2 Coverage of the state of the art requirements for the monitoring architecture by the literature in this context.**

| Platform | Processing | | | Data acquisition | | Scalability | Elasticity | Fault tolerance |
|---|---|---|---|---|---|---|---|---|
| | Parallel | Concurrent | Distributed | Publisher | Polling | | | |
| BlueDove | | | ✓ | ✓ | | ✓ | ✓ | ✓ |
| SEMAS | | | ✓ | ✓ | | ✓ | ✓ | ✓ |
| POLDERCAST | | | ✓ | ✓ | | ✓ | | ✓ |
| Kafka | ✓ | | ✓ | ✓ | | ✓ | | ✓ |
| STREAMHUB | ✓ | ✓ | ✓ | ✓ | | ✓ | | |
| E-STREAMHUB | ✓ | ✓ | ✓ | ✓ | | ✓ | ✓ | ✓ |
| SREM | | | ✓ | ✓ | | ✓ | | ✓ |
| SPS | ✓ | ✓ | ✓ | ✓ | | ✓ | | |
| MonARCh | ✓ | ✓ | ✓ | | ✓ | ✓ | | ✓ |

requirements listed in "Requirements of the Linked Data Monitoring Architecture", as shown in Table 1.

Linked Data Notifications (LDN) is another push/pull protocol (*Capadisli et al., 2017*) and a W3C recommendation (*Capadisli & Guy, 2016*). It defines the way that the linked data notifications are shared and reused among applications. There are three parties in this protocol: senders, receivers, and consumers. Senders produce and send the notifications to the receivers that act as inboxes. Consumers advertise where their receivers are and pull the data from them. It has several implementations like dokieli, sloph, solid-notifications, Virtuoso+ ODS-Briefcase, *etc.*, (*Capadisli & Guy, 2016*). LDN has the ability to work on the web of data. It focuses on the dissemination and decentralization of notifications. However, it has no monitoring feature for the SPARQL queries. Also because LDN is a protocol rather than a system, it is not considered as scalable (See in the Table 1).

## Publish/subscribe relevant literature
Unlike the systems discussed in the "Linked Data Relevant Literature", state of the art publish/subscribe systems have excellent features and techniques for delivering relevant content to the consumer. There are two approaches available in the literature as the basis of these systems: attribute (topic) based and content based. Coverage of the state of the art requirements by the literature is shown in Table 2. According to this table, a state of the art system is evaluated under the following metrics:

- *processing* ability: way of working of its components
  - *parallel*: Dividing a task into multiple subtasks and processing them at the same time
  - *concurrent*: Processing different tasks at the same time
  - *distributed*: Processing tasks using resources of the multiple communicating machines *via* message passing

- *data acquision*: The technique for retrieving the new data from the data sources

- *scalability*: The ability to increase resources and the processing ability for adapting to the increasing service demands
- *elasticity*: Adapting (increasing or decreasing) its resources and the processing ability to the changing workloads
- *fault tolerance*: The level of ability that the system to recover from error conditions

Topic-based approach narrows the search space. Hence the systems following this approach are both scalable and fast. BlueDove (*Li et al., 2011*) and SEMAS (*Ma et al., 2014*) (inspired by BlueDove) are such systems designed to work in the cloud computing environment. BlueDove has been implemented by extending the source code of Cassandra (https://cassandra.apache.org) excluding the storage parts. To handle scalability, the system employs a gossip-based network on the hardware level. To match messages with the appropriate server, it utilises a multidimensional partitioning technique at the software level. SEMAS uses a one-hop lookup overlay to reduce clustering latency. Both systems also examine the data skew and adapt to changes in the burst of event workloads, allowing them to be elastic. Except for the ability to process as parallel and concurrent, they cover all requirements listed in the Table 2. In contrast, instead of a cloud computing environment POLDERCAST (*Setty et al., 2012*) runs over a scalable computer network overlay aiming for fast dissemination of the topics. The novel aspect of this work is that it sends messages in a scalable manner using the network's rings, proximity, and cyclon modules. As one of the most widely used and well-known works Kafka (https://kafka.apache.org) (*Kreps, Narkhede & Rao, 2011*) is a publish/subscribe system and a distributed streaming platform. A "topic" is the collection of a specific stream of records. These published messages and/or streams are then stored in "brokers" which are actually Kafka clusters made up of a collection of server nodes. Consumers can subscribe to the topics and receive data streams from brokers. Kafka is so scalable and fault tolerant that its performance far exceeds two implementations of the popular topic based MQTT protocol (*Yassein et al., 2017*; *Bender et al., 2021*) for IoT systems (ActiveMQ (http://activemq.apache.org) (*Snyder, Bosnanac & Davies, 2011*) and RabbitMQ (https://www.rabbitmq.com)), according to the evaluation results. POLDERCAST and Kafka are not categorized as cloud-based approaches in their work, but both can easily be adapted to work in a cloud environment. While Kafka can divide the message consuming task into subtasks for parallel processing, POLDERCAST does not have this feature. Furthermore, neither of them can process its various tasks concurrently nor is elastic on its own, but Kafka allows for elasticity and concurrency to be implemented.

The content-based approach, on the other hand, emphasizes expressiveness and broadens the term scope. Similarity metrics and techniques are used in event matching algorithms to determine how similar content is to a published message. This can result in a system slowdown as a scalability trade-off. STREAMHUB (*Barazzutti et al., 2013*) is a scalable content-based approach aimed at high throughput that runs on a public cloud environment. The publish/subscribe engine is divided into many logical and operational pieces in this work to support parallel execution in cloud environments. Following that, STREAMHUB is modified to become elastic, as E-STREAMHUB (*Barazzutti et al., 2014*)

in another study. E-STREAMHUB, as a state of the art pub/sub system, meets all of the requirements in Table 2. The system evaluation results show that it can react and adapt dynamically to changing workloads. Another cloud and content-based pub/sub system is SREM (*Ma, Wang & Pei, 2014*) aiming to maximize matching throughput under a big load of subscription requests. It is a distributed overlay SkipCloud using a prefix routing algorithm to achieve low latency. As can be seen in Table 2 while SREM has the basic functionalities, it does not have features such as parallelism, concurrency and elasticity which put it forward from the other state of the art systems.

Although all the systems mentioned so far meet most of the state of the art requirements in Table 2, they are not designed to monitor SPARQL queries locally and over the Web of Data. As a result, with the exception of working as distributed and being scalable, they fail to meet any of the requirements listed in "Requirements of the Linked Data Monitoring Architecture" (*Roffia et al., 2016*) on the other hand, present a semantic pub/sub architecture called SPS to support smart space applications in IoT. SPARQL update queries are used by publishers to make changes to the endpoint. A novel algorithm in the SPARQL subscription (SUB) engine checks query results from a key-value lookup table to detect events. Despite the fact that the system is scalable and works in parallel, earning it a place among the other state-of-the-art systems in Table 2, it can only monitor basic (non-federated) SPARQL queries and lacks the ability to work on the Web of Data, as shown in Table 1.

## REQUIREMENTS OF THE LINKED DATA MONITORING ARCHITECTURE

During design science research, the design problem which outlines the research purpose should be defined using the template given below (*Wieringa, 2014*):

- improve <a problem context>
- by <(re)designing an artifact>
- that satisfies <some requirements>
- in order to <help stakeholders achieve some goals>

When this template is applied to the linked data monitoring problem, the following design problem definition which is compatible with the requirements listed in *Umbrich, Villazón-Terrazas & Hausenblas (2010)*, is formed:

- improve linked data change monitoring capability of applications
- by designing a scalable linked data query monitoring architecture
- that satisfies the following requirements:

  1. System should isolate developers from the details of the Linked Data monitoring by providing monitoring as a service.
  2. In addition to working with the local RDF datasets, the system should be able to work with multiple remote datasets on the Web of Data.

3. Other systems should be able to monitor just the information they require by registering single, raw or federated SPARQL queries to the designed monitoring system.

4. When a change occurs in the dataset relevant to a SPARQL query result, the system should detect this change after the occurrence.

5. System should communicate the SPARQL result changes to the registered clients after the detection preferably with a mechanism similar to the publish/subscribe.

6. The system should convert raw SPARQL queries into the federated format.

7. The system should be distributed and scalable in terms of the number of active nodes in the cluster. As the number of query registration requests increases, so does the amount of data handled by the system, and the number of client applications. Furthermore, increasingly complicated inquiries necessitate monitoring an increased number of simple SPARQL queries. Thus, in all of these scenarios, the system must remain operational and handle requests smoothly and effortlessly.

- in order to:

  - enhance the response-ability of client applications to data changes so that end-users can be notified on time,
  - isolate developers from the details of linked data monitoring.

After this step, knowledge questions should be defined. These questions aid in exploring the artifact to be designed in terms of its context and its relationship with the context. Due to space limitations, only the knowledge questions used for the evaluation and validation of the architecture are given and discussed in "Evaluation and Validation of the Architecture".

## DESIGN OF THE ABSTRACT ARCHITECTURE AND THE VALIDATION MODEL

According to the methodology of Design Science, the artifact that is to be designed and its relevant context should be investigated to form a conceptual framework. The conceptual framework consists of named constructs that explain the structure of context and artifact. Constructs of the conceptual framework fall into two categories: architectural and statistical structures.

The problem domain for the linked data monitoring artifact is linked data space. Therefore, architectural structures of the conceptual framework are *Linked Data space*, *SPARQL endpoints*, *SPARQL queries*, *federated SPARQL queries*, *join algorithms* for joining the results of single SPARQL queries, *a DaDy definition* which defines the frequency of change for each dataset, *a push/pull mechanism* since data push is not mandatory for SPARQL endpoints, *system components implemented with the actor model* to provide scalability, *cluster dynamics* for scalability.

A statistical structure is a population of elements and several attributes of them with a probability distribution within the population (*Wieringa, 2014*). In other words, statistical

structures are variables which can be used in the definition of phenomena in a treatment. For the linked data monitoring artifact they are the change dynamics and DaDy definitions of SPARQL endpoints.

Design Science Methodology necessitates an empirical cycle (*Wieringa, 2014*) to validate and evaluate the implementation of the abstract architecture. Following this, we built a validation model that includes a representative sample of our proposed architecture as well as the anticipated context for it. In addition, we developed validation and knowledge questions to guide us through the evaluation and validation of our implemented architecture. Details of the empirical process, the guiding questions and the inferences from the collected data during these experiments are examined in "Evaluation and Validation of the Architecture".

The purpose of the Linked Data monitoring architecture is to take the periodic data monitoring load from the programmer and the application he/she develops by providing Linked Data monitoring as a service. Therefore it should satisfy the following "*as a service*" requirements:

- The system should be able to monitor the results of federated SPARQL queries at the change frequencies of the SPARQL endpoints involved.
- Diverse client applications should be able to register new federated SPARQL queries for monitoring.
- For each query, the system should be able to interact dynamically with SPARQL endpoints which is a unique feature of the system.

These requirements make the presence of the following context assumptions necessary:

- To define the periodic querying frequency of the data sources, the DaDy metadata of related SPARQL endpoints should be present.
- To monitor data sources continuously, there should be no network problems.
- To monitor data sources continuously, SPARQL endpoints should be operational all the time.

The combination of these requirements, assuming the context assumptions are met, is expected to benefit the programmer by isolating her/him and her/his code from the complexities of Linked Data monitoring.

Because the proposed architecture necessitates a high level of concurrency, the validation model was designed using the asynchronous actor model (*Hewitt, Bishop & Steiger, 1973*; *Hewitt, 2010*). Subscribers send SPARQL queries to the monitoring engine about their interests to be notified of any future changes. The monitoring engine recognizes query and endpoint pairs and distributes execution work across the cluster. Worker actors do the heavy work by scheduling themselves to run a query over its relevant endpoint regularly. New query results are compared to previous ones to see if there are any differences.

MonARCh can monitor both simple and federated SPARQL queries. In the federated case, detecting changes in query results necessitates recalculating the SERVICE clause

results *via* join operations. For the generation of federated query results, the hash join technique is used to adapt to the distributed and concurrent nature of the actor model. Because the SERVICE clause results are disjoint in this technique, simultaneous join operations can be performed concurrently. Actors perform join operations using a hash join algorithm designed for parallel multiprocessor architectures such as the GRACE (*Kitsuregawa, Tanaka & Moto-Oka, 1983*) and the Hybrid (*DeWitt et al., 1984*). All actors are deployed automatically on a computer cluster. A modulo-based simple routing algorithm can be used to distribute actors on the cluster.

From the architectural point of view Comunica (https://comunica.dev; *Taelman et al., 2018*), a modular meta query engine and knowledge graph querying framework, is similar to MonARCh. It is written in Javascript to make it web-friendly and modular. It is neither a publish/subscribe architecture nor a linked data monitoring architecture. It can be used for many purposes like federated query processing or customized RDF parsing depending on the modules set up and their configuration in a specific Comunica instance. Additionally, it is built on top of the actor model, similar to MonARCh. Furthermore, it employs the publish/subscribe design pattern to decouple its modules from one another and the mediator software pattern to connect the relevant decoupled blocks when necessary. As a result, Comunica, as an innovative work with the actor model at its core, encourages the solidity and validity of the design choices in our architecture.

The abstract architecture of MonARCh is depicted in Fig. 1. Federator, Distributor, Executor, Parallel Join Manager, and Hash Join Performer are the five actor-based components. Federator resides at the top of the actor hierarchy with critical roles such as distributing federated query across Distributors, collecting SERVICE clause results, keeping track the change of federated query result, initiating and controlling the join process which makes it manager of the query monitoring pipleline. A SERVICE clause which resides in a federated query may possibly executed over multiple endpoint. For this reason Distributor serves as an auxiliary component that holds SPARQL endpoint list to distribute the relevant SERVICE clause over and execute on each of them. It then collects and merges the results of this SERVICE clause from the relevant endpoints. Thirdly, Executor is an atomic and core component residing at the bottom of the actor hierarchy which is responsible to execute a SERVICE clause over its assigned endpoint across its update intervals and keeps track the result changes. On the other hand, ParallelJoinManager resides at the heart of the architecture which is responsible to divide, parallelise and manage the GRACE hash join operation when a join task of two SERVICE clauses is issued by the Federator. Finally, a HashJoinPerformer is an ant like worker component which accomplishes the GRACE hash join operation of the divided result pieces of the SERVICE clauses assigned by the ParallelJoinManager. While the Federator, Distributor, and Executor manage the query monitoring flow, they must be persistent and have their own regions. These regions control the creation of appropriate actor instances and the message routing between them. Parallel Join Manager and Hash Join Performer, on the other hand, are primarily result-oriented. Their outcomes may differ and may change over time. As a result, these components must be recreated for each join operation and should not be persistent. If the query is federated, the Federator splits it into SERVICE clauses. For each

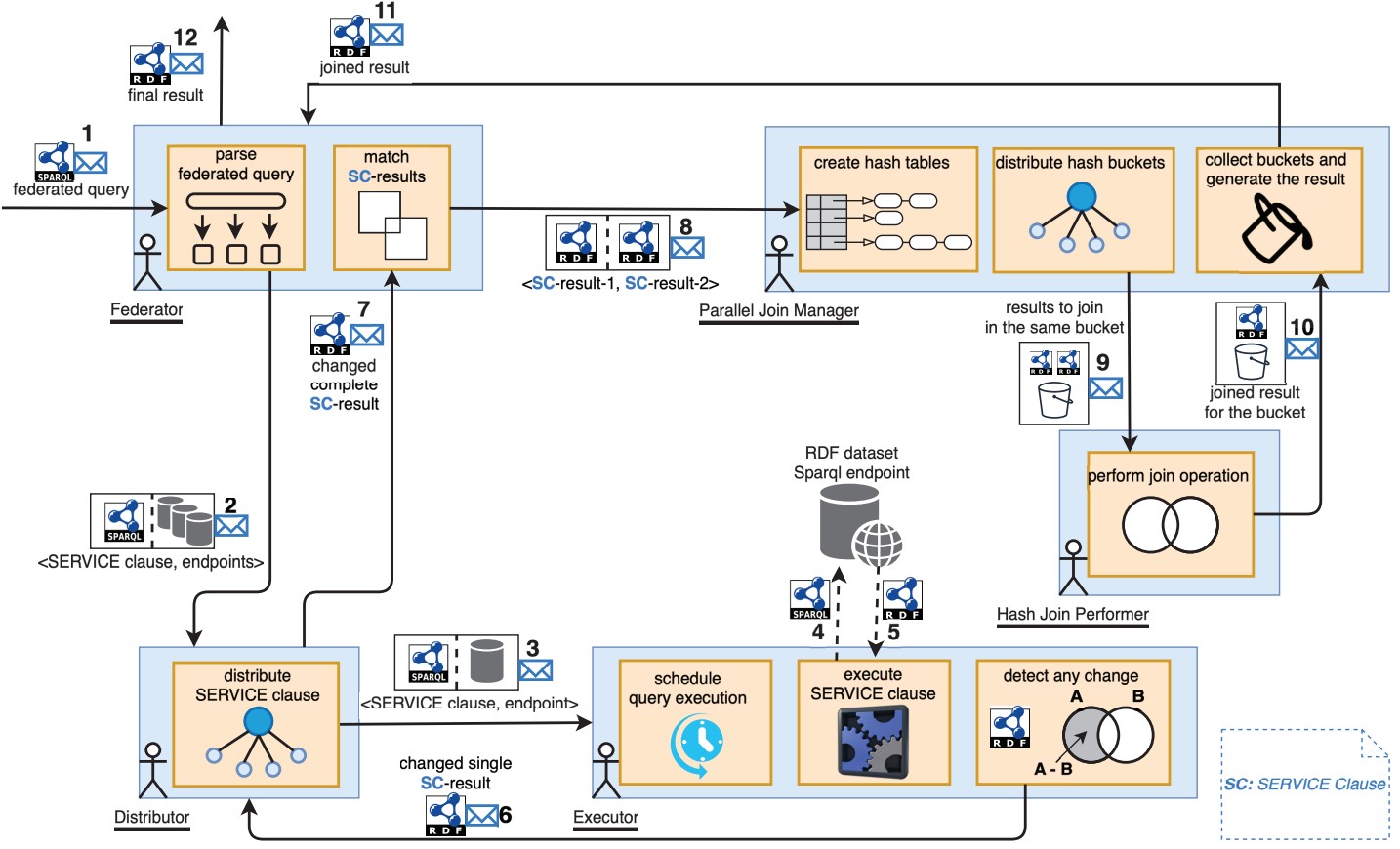

**Figure 1 Query monitoring pipeline of MonARCh.** RDF logo: https://cygri.github.io/rdf-logos/svg/rdf.svg, SPARQL logo: https://cygri.github.io/rdf-logos/svg/sparql.svg. Note: The figure has been drawn in https://draw.io using the components supported by the service itself except for RDF and SPARQL logos. Created in draw.io; RDF icon and SPARQL logo source credit: http://www.w3.org/RDF.

detected SERVICE clause, relevant SPARQL endpoints are assigned. The <SERVICE clause, endpoints> pairs are then sent to the Distributor region, which delivers messages to the appropriate Distributor actors (step 2 in Fig. 1). The Federator is also in charge of keeping track of the results for the SERVICE clauses that will be delivered by Distributors. Because these SERVICE clause results are then sent to the Parallel Join Manager, which joins them into a single main result (steps 8 and 9 in Fig. 1) using the novel Parallel GRACE Hash join algorithm which has been elaborated in "Parallel GRACE Hash Join Algorithm". Similarly, when the Distributor receives a message with the <SERVICE clause, endpoints> pair, it sends the SERVICE clause to the relevant Executor actor for each endpoint *via* the Executor region (step 3 in Fig. 1), then collects and merges the results. If the SPARQL endpoint for executing a SERVICE clause is not operational, the Executor returns the last cached result. Finally, when the Executor receives a message containing a <SERVICE clause, endpoint> pair, it executes the SERVICE clause against the relevant endpoint. This module is the key component that attempts to detect changes in the outcome of a SERVICE clause. It schedules itself to re-execute the query and obtain the

---

**Algorithm 1** Parallel GRACE hash join algorithm in ParallelJoinManager[Actor].

1: $m \Leftarrow number\_of\_buckets$

2: $w \Leftarrow 0$

3: **for each** $r \in \mathscr{R}$ **do**

4:    $i \Leftarrow h(r.a) \bmod m$

5:    $bucket_i \Leftarrow bucketMap_R[i]$

6:    $bucket_i \Leftarrow bucket_i \cup r$

7:    $bucketMap_R[i] \Leftarrow bucket_i$

8: **end for**

9: **for each** $q \in \mathscr{Q}$ **do**

10:    $j \Leftarrow h(q.b) \bmod m$

11:    $bucket_j \Leftarrow bucketMap_Q[j]$

12:    $bucket_j \Leftarrow bucket_j \cup q$

13:    $bucketMap_Q[j] \Leftarrow bucket_j$

14: **end for**

15: $bucket_R \Leftarrow POP(bucketMap_R)$

16: $bucket_Q \Leftarrow POP(bucketMap_Q)$

17: **while** $bucket_R \neq EoR_R$ and $bucket_Q \neq EoR_Q$ **do**

18:    $actor_{HJP} \Leftarrow new\ HashJoinPerformer^{Actor}$

19:    $message_{PHJ} \Leftarrow new\ PerformHashJoin(bucket_R,\ bucket_Q)$

20:    $send(message_{PHJ},\ actor_{HJP})$

21:    $w \Leftarrow w + 1$

22:    $bucket_R \Leftarrow POP(bucketMap_R)$

23:    $bucket_Q \Leftarrow POP(bucketMap_Q)$

24: **end while**

---

new results based on the estimated change interval of the endpoint. The new results are compared to the old ones to see if there are any differences. When a change is detected, it is communicated to the Distributor. The Distributor gathers the new results of its relevant SERVICE clause, which may be executed over multiple endpoints. It returns to the Federator with new merged results for the SERVICE clause. When the Federator receives a new (changed) result, it searches for a matching SERVICE clause result using a common query variable. Two SERVICE clause results containing matching common query variable is then sent to the Parallel Join Manager for the join plan. Parallel Join Manager generates hash tables for both results using the Grace Hash Join algorithm according to a fixed bucket size. Buckets of the hash maps are paired according to keys for parallel execution of the join operation, and each pair is sent to a separate Hash Join Performer actor (See Algorithm 1). The Hash Join Performer performs a simple hash join operation and returns the joined results to the Parallel Join Manager (See Algorithm 2). Parallel Join Manager

---

**Algorithm 2** Performing hash join operation in HashJoinPerformer[Actor].

1: **for each** $r \in bucket_R$ **do**
2:    $i \Leftarrow h(r.a)$
3:    $R_i \Leftarrow hashMap_R[i]$
4:    $R_i \Leftarrow R_i \cup r$
5:    $hashMap_R[i] \Leftarrow R_i$
6: **end for**
7: **for each** $q \in bucket_Q$ **do**
8:    $j \Leftarrow h(q.b)$
9:    $Q_j \Leftarrow hashMap_Q[j]$
10:    $Q_j \Leftarrow Q_j \cup q$
11:    $hashMap_Q[j] \Leftarrow Q_j$
12: **end for**
13: $Result \Leftarrow \varnothing$
14: **for each** $i \in hashMap_R$ **do**
15:    **if** $Q_i \notin \varnothing$ **then**
16:      $Q_j \Leftarrow Q_j \cup q$
17:      $Result \Leftarrow Result \cup (R_i \bowtie Q_i)$
18:    **end if**
19: **end for**

---

---

**Algorithm 3** Constructing final join result in ParallelJoinManager[Actor].

1: $w \Leftarrow w - 1$
2: $FinalResult \Leftarrow FinalResult \cup SubResult$
3: **if** $w = 0$ **then**
4:    **return** $FinalResult$
5: **end if**

---

gathers joined results from all buckets, merges them, and produces the final joined results (See Algorithm 3). The Federator receives the final results. When the Federator receives the final results for the first time or as a change, it sends the results to the issuer agent of the main SPARQL query.

## Parallel GRACE hash join algorithm

We have built a novel parallel implementation of the GRACE hash join algorithm which is depicted in the Algorithms 1–3 respectively in three steps. *ParallelJoinManager[Actor]* is the key component of the system which manages parallelizing the whole join process.

Therefore, firstly in Algorithm 1 carried out by *ParallelJoinManager*[^Actor], for the parallel join execution of the results $R$ and $Q$, they both are distributed into the fixed number of buckets (steps 3 to 8 and 9 to 14). Buckets contain pieces of the results which are logically grouped to be joined parallel. The most important point of the parallelizing operation is finding the join attribute (or attributes) and revealing the attribute values of the relevant result tuples.

Bucket number $i$ and $j$ for each tuple of the results $R$ and $Q$ are calculated respectively by processing the hash code for the join attribute value of each tuple into modulo operation with the decided number of buckets. For tuples to be joined $r$ and $q$ would have the same join attribute value. Thus in this situation, $r.a$ equals $q.b$ and they produce exactly the same hash code and bucket number. Therefore this operation guarantees that grouping the tuples to be joined under the same logically numbered buckets.

Each bucket of the results $R$ and $Q$ are stored in the respective hash maps under the calculated bucket number. Both hash maps are equal in size with a constant number of buckets. For each bucket of the hash maps $R$ and $Q$ with the same order (under the same key) are the only candidates to be joined because of the logical grouping. For this reason a new *HashJoinPerformer*[^Actor] has been created and a new *PerformHashJoin* message containing the buckets to be joined is sent to this actor. Finally, the waiting join count $w$ is increased to track if all parallel join operations have been performed to generate the final join result thereafter. After all buckets of the hash maps $R$ and $Q$ have been processed in this way, parallelizing the hash join operation has been completed.

Performing the hash join operation for the relevant buckets carried out by the *HashJoinPerformer*[^Actor] has been depicted in the Algorithm 2. It has two main phases as *build* and *probe*. In the build phase (steps 1 to 6 and 7 to 12), hash maps for both buckets are created. For each tuple in the bucket $R$ and $Q$, the hash code of the join attribute value has been calculated. Then, each tuple has been added to the result set relevant with this hash code and they are stored in the respective hash map for this bucket. After both hash maps have been built, the probing operation is iterated over the smaller map (steps 13 to 19). Therefore in the algorithm, we took the hash map for bucket $R$ into the account as the smaller one. Each key $i$ of the smaller hash map $R$ is iterated and if the result is non-empty for that key in hash map $Q$, then the join value of the $R_i$ and $Q_i$ is added to the final result.

Finally, after the join operation has been distributed over the *HashJoinPerformer*[^Actor]s for the parallel execution, joined results are collected by *ParallelJoinManager*[^Actor] as can be seen in the Algorithm 3. When a joined result has been received by the *ParallelJoinManager*[^Actor], waiting to join count $w$ is decreased and the received sub result is added to the final join result. When there is no more join operation to be waited for, this means the parallel hash join operation has been completed, therefore the final result is sent to the *Federator*[^Actor].

## IMPLEMENTATION OF THE VALIDATION MODEL

For building the monitoring architecture and managing the cluster, we used the Akka (https://akka.io) toolkit (*Gupta, 2012*) as the actor model implementation on JVM (Java Virtual Machine) in the prototype system. We chose Scala (https://www.scala-lang.org) as the implementation language because it has built-in concurrency management and is also

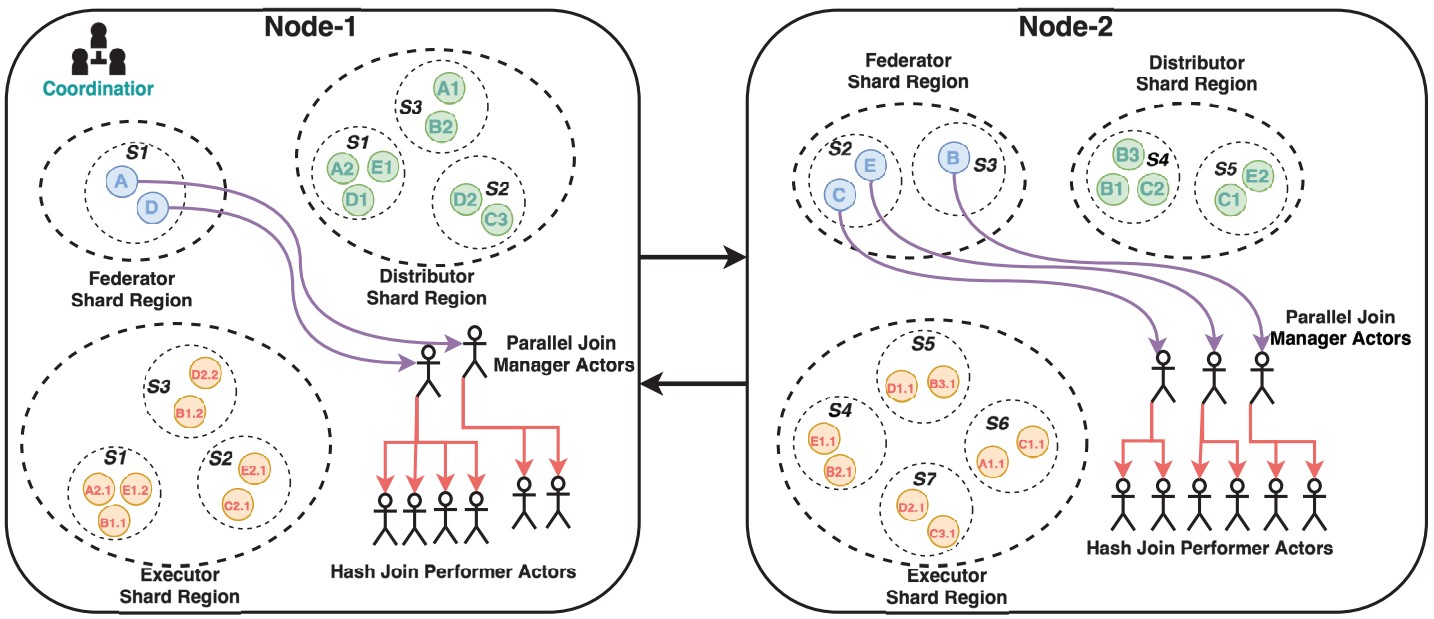

**Figure 2** Sample cluster structure model for MonARCh. Note: The figure has been drawn in https://draw.io using the components supported by the service itself. Created in draw.io; RDF icon and SPARQL logo source credit: http://www.w3.org/RDF.

Akka's implementation language. The system is highly scalable and concurrent due to Akka's cluster sharding property. Furthermore, it has high performance and is resilient to the burst of message workloads, which can reach thousands of CPU and memory bound actors and tens of thousands of msg/sec per computer cluster node. From the application point of view big enterprises such as Tesla, PayPal, LinkedIn, Starbucks *etc.*, have relied on using Akka and actor model on production. For example Tesla has built its virtual power plant architecture (https://www.infoq.com/presentations/tesla-vpp) as an IoT application upon Akka. On the other hand PayPal has also rebuilt its web crawler which resides at the heart of their big data platform using Akka. They could achieve billions of daily transactions (https://www.lightbend.com/case-studies/paypal-blows-past-1-billion-transactions-per-day-using-just-8-vms-and-akka-scala-kafka-and-akka-streams), 90% CPU utilization rate and 80% reduction of their code base. When a small memory footprint has been used through the system Akka argues that it can support up to 200 millions msg/sec on a single machine and up to 2.5 million actors per GB of heap.

Figure 2 shows a sample MonARCh cluster built with the Akka cluster sharding feature. *Helland*'s *(2007)* work inspired the roots of Akka's cluster sharding architecture. Messages are used to send query results between actors in this architecture. Since some results may be too large, serialization and transmission *via* TCP communication between actors running on different nodes may be difficult. As a result, UDP messaging configuration is used, which is better suited for time-sensitive applications and can easily transmit larger messages. Akka uses Artery Aeron UDP (https://akka.io/blog/article/2016/12/05/aeron-in-artery) driver for transmitting remote actor messages through the cluster with high

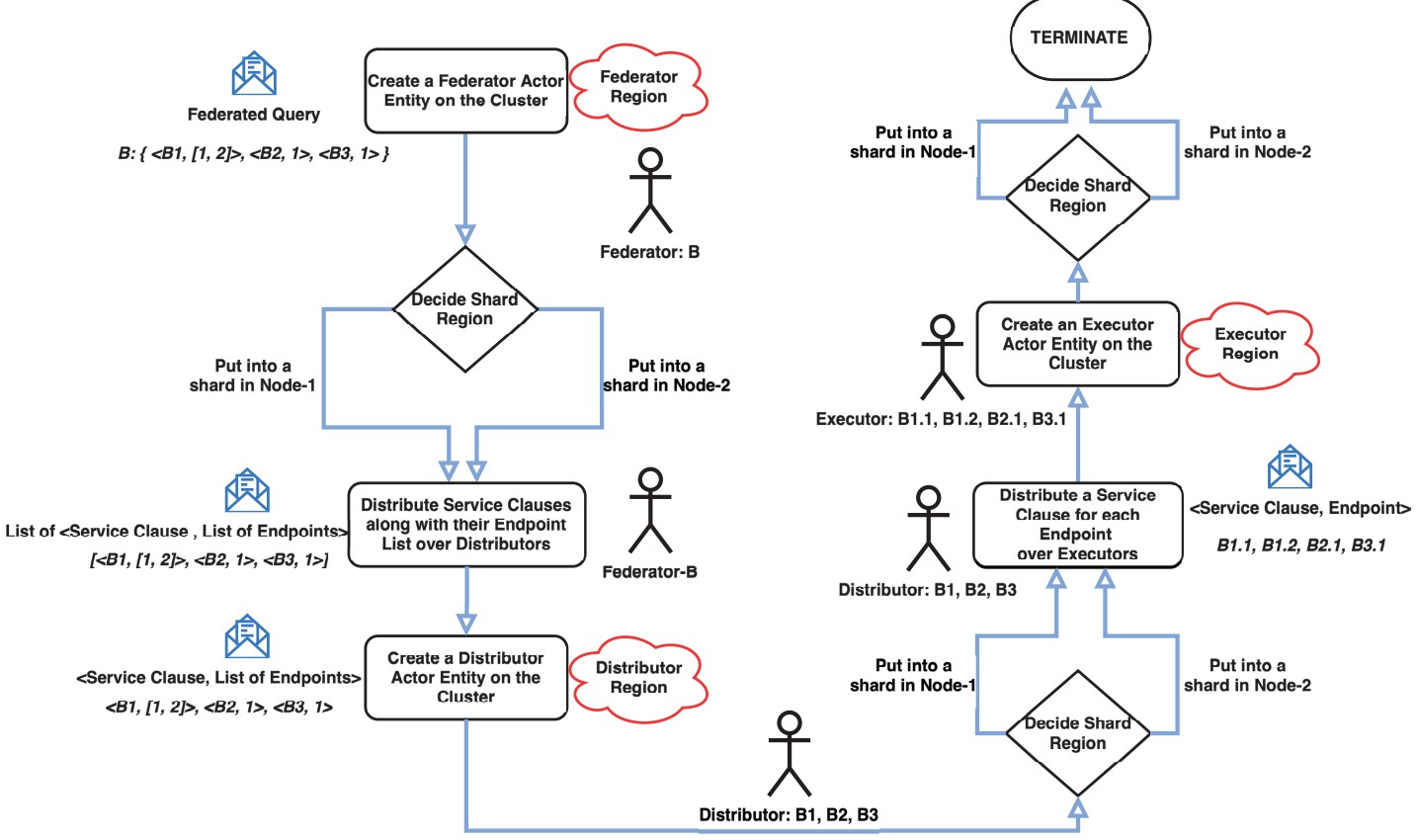

**Figure 3  Message flow and actor interactions in the cluster.** Note: The figure has been drawn in https://draw.io using the components supported by the service itself.                                                  

throughput. It is designed to deliver 1 million messages/sec with a latency of 100 microseconds. It is based on busy spinning that a message is tried to be sent continuously until it is successful. Thus, even if it is based on UDP it provides as close delivery rates as TCP. Artery Aeron UDP (https://doc.akka.io/docs/akka/current/remoting-artery.html) comes as a solution when the system requires high throughput and low latency as MonARCh. We also moved from TCP to UDP when maximum data frame size exceeds 8 MB.

The cluster has a coordinator actor who routes incoming messages to the appropriate node using the gossip protocol. The implemented system's three critical modules are the Federator, Distributor, and Executor actor regions. These modules have parent-child relationships and control the creation of entity actors and message routing. Shards are simply groups of actor instances, named entities that reside in regions. Thus, this structure makes it easier to move these entities later on if necessary. Because the previously mentioned cluster architecture provides location transparency for actors, it is unnecessary to know which actor is running on which node. Messages are simply sent to any node's region, and cluster management handles the rest, including load balancing for actor

deployment and message delivery. Entity actors are depicted as letters (some with numbers) with colored circles arranged in shards coded with the letter S and shard number. Each cluster node has its own shard region, and shards are distributed across the cluster based on the same logical region. There are three shard regions named Federator, Distributor and Executor. These regions are owned by both nodes but managed as the cluster shown in Fig. 2. A federated query coded with the letter B and represented as a blue circle, for example, resides in Node-2's Federator shard region. Its SERVICE clauses with relevant endpoints have the same letter and query number as B1, B2, and B3, which are represented as green circles in the Distributor shard region. While entities may be located in the same region logically, they may be located in different nodes. B2 is in Node-1, while B1 and B3 are in Node-2. In the Executor shard region, SERVICE clauses with relevant endpoints are similarly coded with the same "SERVICE clause code", "a dot", and "an endpoint number" as B1.1, B1.2, B2.1, and B2.3. Furthermore, for clarifying this concepts a flowchart which depicts the message flow and actor interactions in the cluster has given in Fig. 3. Because the actors of Parallel Join Manager and Hash Join Performer are not managed by a cluster, they exist in the same node as their parent actors. They are both join relevant actors and a join operation with the same two results is only executed once and never used again. Therefore, these actors are destroyed once the join operation is completed. Federator creates Parallel Join Manager, and Parallel Join Manager actors create Hash Join Performer. Source code of MonARCh can be accessed *via* the following URLs: https://github.com/seagent/monarch and https://doi.org/10.5281/zenodo. 10679208.

## EVALUATION AND VALIDATION OF THE ARCHITECTURE

According to the Design Science Methodology, before an evaluation and validation study begins, knowledge questions that serve as a guide during the study should be prepared. Therefore, before discussing the evaluation and validation studies conducted, it is necessary to give the prepared questions.

**Requirements satisfaction questions**

For this reason, we have built the requirements satisfaction questions defined for Linked Data monitoring architecture which are given below.

1. *Does the artifact isolate developers from the details of Linked Data monitoring? (What assumptions does the artifact make about the Linked Data monitoring effort required by the developer?)*
2. *How much time passes between the detection of a query result change and the notification of the registered applications? Does it satisfy functional requirements?*
3. *How much time passes between query result change and change detection? Does it satisfy functional requirements?*

To be able to answer the requirements satisfaction questions, the system should be evaluated. Knowledge questions that guided the evaluation process in this study are given below. They are logically grouped under two categories that "Performance Analysis" and "Design Discussions".

**Performance analysis questions**

In order to be accounted as state of the art, a SPARQL monitoring architecture need to cover both functional (linked data specific) and non functional (state of the art) requirements listed in Tables 1 and 2 respectively. In the light of these requirements combined; performance, scalability and limits of the system are investigated under the performance analysis questions given below.

1. *Was the expected number of notifications gotten?*
2. *What was the maximum number of observed federated queries?*
3. *How many queries per unit of time could be successfully registered?*
4. *What limits the number of queries being monitored?*
5. *What limits the number of queries that are registered per unit of time?*
6. *How does the number of SERVICE clauses in a federated query affects the performance of the system?*
7. *What happens if the query selectivities change?*
8. *What happens if query registration requests get more frequent?*
9. *What are the network limits of the system?*
10. *Which factors affect the usage of the system resources?*

**Design discussions questions**

In the abstract architecture, we chose the actor model for scalability and concurrency due to the following advantages:

- Each actor encapsulates its state and behavior, and communicates with other actors solely through message passing. This isolation prevents shared memory issues like race conditions and deadlocks.
- Actors communicate *via* messages, allowing for concurrency without the complexity of shared state management. This simplifies concurrent programming and reduces the likelihood of bugs related to shared state.
- Actors can be distributed across multiple machines without requiring changes to the internal logic of the actors themselves. This scalability makes it suitable for building large-scale, distributed applications.
- In the Actor model, each actor operates independently, and failures are isolated to individual actors. Supervision hierarchies can be used to manage actor failures, allowing systems to recover from errors gracefully without affecting the overall system stability.

Additionally, a parallel hash join algorithm was chosen since it supports the concurrency and scalability provided by actor models.

Finally, to notify registered applications of the changes in the results of the federated SPARQL queries on time, it is necessary to know the change frequencies of the datasets connected to the SPARQL endpoints involved. However, these definitions, which are embedded in the VoID definitions of SPARQL endpoints as DaDY definitions, may not be provided by the relevant SPARQL endpoints.

With all these points in mind, it is necessary to answer the following design discussion questions to validate the proposed abstract architecture:

1. *How would the artifact be affected if a different join algorithm was used?*
2. *How would the artifact be affected if the actor model was not used?*
3. *What happens if a query should be answered by a SPARQL endpoint with no DaDy definition?*
4. *What happens if a query should be answered by a SPARQL endpoint with no VoID definition?*

After devising the knowledge and requirements satisfaction questions, experiments were conducted. In the following subsections, these experiments, the data acquired during them and inferences made with the guidance of the knowledge and validation questions are examined.

Evaluation section is organized as follows. First, "Experimental Settings" gives details about the experimental environment and settings, on the other hand "Evaluation Story and Test Scenarios" dives into the pros and cons of the alternative benchmarks while elaborating our unique dynamic test benchmark. While "Execution of the Evaluation" explains the execution details of the evaluation such as components, setup, measurement and data collection, finally "Data Analysis" gives thorough analysis about the evaluation results including interpretation of graphics, answering knowledge questions and comparison to similar systems.

## Experimental settings

The prototype for MonARCh was run on a two-node cluster. Each node on the cluster had Intel Xeon E5 2620 v4 (8 core-16 thread) processor, 32 GB RDIMM RAM, 1.8 TB 15K SAS Disk with RAID 5 configuration and Ubuntu 18.04 operating system installed. The Akka toolkit which is an actor model implementation on JVM has a cluster sharding feature that lets the system work as a cluster on multiple nodes (computers) and provides location transparency for actors enabling easy access to them. Therefore the actor system can make use of multiple JVMs running on different nodes. Akka is one of the most modern, robust and mature actor model frameworks. It has an excellent documentation and broad community which lets easy to use with lots of examples. Moreover from the familiarity and flexibility point of view, it has programming language support for both Scala and Java. Overall, Akka actor framework provides a powerful and flexible platform for building concurrent, distributed, and resilient applications, making it well-suited for a wide range of use cases in modern software development.

The context in which the system operates (that is the SPARQL endpoints) was designed as a closed world in the laboratory so that it is isolated from any outside events that can influence the measurements taken. This allowed measuring the performance of the system reliably, without the need to consider any Internet and network problems. As the RDF server and SPARQL endpoint, we preferred Virtuoso (https://virtuoso.openlinksw.com) which serves also as a data management platform. It has well documentation and easy to

configure *via* its management dashboard. Two nodes in the cluster had Virtuoso (*Erling & Mikhailov, 2010*; *Erling, 2012*) RDF servers installed and running on them that serve as the SPARQL endpoints for DBPedia, NyTimes and Stockmarket datasets. The Stockmarket dataset is a synthetically generated dataset, DBpedia and NyTimes datasets are taken from FedBench suite (*Schmidt et al., 2011*) enhanced with the synthetic data suitable for the evaluation story.

Virtuoso services were deployed with Docker (https://www.docker.com) configuration on three separate nodes. Docker offers numerous advantages for modern software development and operations, including consistency, isolation, portability, efficiency, flexibility, scalability, and integration with DevOps practices. These benefits make Docker a popular choice for building, deploying, and managing containerized applications in a variety of environments. Therefore RDF server deployment becomes portable and is kept in isolated from other services alongside the server nodes. The actor cluster was built upon the first and second nodes and queries were issued to the system *via* a separate client actor system running on the third node. Each actor system involved in the cluster and the client actor system were configured to keep its statistics in a log file. Redis (https://redis.io) key-value store was installed on the third node to ensure that the actor system functions properly using the observational data collected. Because of MonARCh has been deployed as a cluster running two separate JVMs with interaction, metrics such as total query count and actor count are stored on Redis which serves as a mediator component to track these metrics. The prototype was designed to log data during execution to make the necessary inferences to validate and evaluate the proposed conceptual architecture. These logs contained data related to metrics such as query processing time, change notification time, resource usage related to CPU and memory, and network usage related to message size and count.

## Evaluation story and test scenarios

When evaluating a system which uses SPARQL queries inside its architecture, using a widely accepted benchmark such as FedBench (*Schmidt et al., 2011*) and LargeRDFBench (*Saleem, Hasnain & Ngomo, 2018*), is common sense. Both of these benchmarks have perfectly designed query sets. The queries they include, on the other hand, are more static and can be used to test the performance of federated query engines against static Linked Data from multiple sources. In their current state, the original benchmarks require a significant amount of work and effort to reconfigure for testing the behaviour of a system monitoring the continuously changing results of queries.

Another benchmark that can be thought to be a suitable one is BEAR (*Fernández et al., 2018*). BEAR contains three datasets: A, B, and C. Dataset A consists of the first 58 snapshots of the Linked Data Observatory (*Käfer et al., 2013*). Since we aimed to push the limits of our system, we could not limit the maximum number of queries to 58. Therefore, concerning the number of data changes required for our experiments, this count was insufficient. Dataset B consists of DBPedia Live changesets. This violated our federated query requirement. Dataset C contains 32 snapshots of the European Open Data Portal

(https://data.europa.eu) taken from the Open Data Watch project (https://opendatawatch.com). Like Dataset A, this count was insufficient for our experiments.

Lastly, another dataset that could be used for evaluation is PoDiGG (*Taelman et al., 2017*). PoDiGG could be tailored for our evaluation purposes by using each data file in PoDiGG as the data source for separate SPARQL endpoints. However, instead of tailoring a dataset artificially for our evaluation, we preferred to create a naturally suitable scenario and the necessary datasets for the evaluation of MonARCh.

As a result, rather than modifying FedBench or LargeRDFBench to serve as a dynamic Linked Data monitoring system benchmark, we designed and built a new query set covering the financial domain from the ground up. The query set collects data from multiple dynamic Linked Data sources and is made up of five different types of queries with varying levels of selectivity. Furthermore, each query type, with the exception of the least selective, is a query template in and of itself, allowing the generation of queries with varying levels of selectivity within the context of the template. As a result, it is possible to test the system with a diverse set of queries and through two-stage query selectivity.

The evaluation story follows updates on companies whose data is linked in DBpedia, NyTimes, and Stockmarket datasets. We extended the original schemas of the DBpedia and NyTimes datasets to achieve query diversity, while designing and creating the Stockmarket dataset from scratch.

Because the primary goal of this study is to provide programmers with a service for monitoring Linked Data changes, we attempted to test the limits of our scalable architecture to see how many queries it could monitor at the same time. Since there are only 500+ companies that connect DBpedia and NyTimes, using the datasets in their current form was insufficient for assessing the performance of our system. As a result, we generated 5,000 companies of artificial data for each dataset based on the newly designed Stockmarket schema. We also extended the schemas of DBpedia and NyTimes. While the DBpedia dataset contains some descriptive and statistical information about companies, the NyTimes dataset contains media and news data about companies. The Stockmarket dataset stores financial data about the companies in DBpedia dataset. Technical details about the dataset and template queries are given in Appendix-A (see Appendices given as Supplemental File). The Listing-A1 depicts the schema of datasets related to the financial evaluation story for an example company.

Our evaluation query set is composed of five different query selectivity types which are named least, low, middle, high, and most. These are shown in Listing-A2, A3, A4, A5, A6 respectively.

NyTimes and Stockmarket were updated periodically and continuously by two programs with different periods to create some changes in these datasets. DBPedia dataset was not updated. It was deployed on the same node with the client application from which the query registration requests came. All SPARQL query types displayed in Listing-A2 to Listing-A6 are in a federated form which was converted from raw query form by the SPARQL query federator engine presented in the prototype architecture.

We created test scenarios based on query types and selectivity levels ranging from least to most selective. We also created a template for each query type to provide sub-selectivity

levels by filtering SERVICE clause results. Except for the least selective query scenario, which returns all SERVICE clause results, all others allow filtering SERVICE clause to return 100% to 20% of results by cropping 20% at each sub-selectivity level. It is possible to change the sub-selectivity level of any query type given in Listing-A2 to Listing-A6 by extending or narrowing the range colored in orange. As a result, while the least selective query type has one template, the low, middle, high, and most selective query types each have five templates, for a total of 21 query templates.

## Execution of the evaluation

In the test scenarios, there was a separate actor for each query instance belonging to a designed template that was in charge of registering its dedicated query with the system. These actors acted as MonARCh clients who want to be notified when the results of the queries they register changed. In test scenarios, unique raw queries were created for each generated query template and converted into federated queries using WoDQA (Web of Data Query Analyzer) (*Akar et al., 2012*) federated SPARQL query engine. It analyzes the VoID document metadata of the existing datasets to find and match the relevant triple patterns in the raw query. WoDQA provides an efficient and rule based data source selection mechanism with very low latency which suits well with the architecture of MonARCh requiring high performance. Each federated SPARQL query was sent to the cluster by a different client actor using Akka's cluster client receptionist feature.

Bottlenecks can occur when too many queries are sent at once. Because the system cluster will have to deal with sending too many big sized messages through the network and processing them at the same time. When an actor attempts to send a message, it waits for the UDP driver to become available and transmits all packages of this message. As a result, when the UDP driver is unavailable for an extended period of time while waiting to process the remaining messages due to the current transmission load, the actor system shuts down. Thus queries need to be sent in subsets with a delay by providing sufficient time for messages to be transmitted through the network to the destination actors. This delay allows actors to perform their duties while freeing up operating system resources.

The evaluation was performed iteratively and incrementally using the query templates in the test scenarios. In addition, the total number of query instances sent to the system for each query template increased with each iteration. The current query count was appended at the end of all variables of the relevant query template while looping until the total number of queries was reached. Since unique query instances were generated in this way, each query was treated as a new query. The stock dataset updater ran every 5 min, while the NyTimes dataset updater ran every 10 min. Iterations took between 30 and 45 min. The test scenario for this selectivity level was terminated when the system reached its monitoring boundary for the query count of the current iteration. Measurements of the Akka cluster's query processing performance, system resource usage, and network cost were saved as records in the system log files. Throughout each iteration, there were separate logs for each node in the cluster and the client. These metrics were saved in the Redis server to double-check the total actor count and registered query count. After all iterations were completed, we thoroughly examined the log files of each iteration by

collecting metric values. The source code of the *dataset updater* can be accessed *via* the following URLs: https://github.com/seagent/datasetupdater and https://doi.org/10.5281/zenodo.10679264.

### Measurement metrics

During the experiment, the following metrics were measured for each iteration of a test scenario with the given query count under the selectivity of a query type:

- **Total** {*Query Count, Query Count per Minute, message size per second sent through the network, message count per second sent through the network, message size sent through the network, message count sent through the network*}
- **Max** {*message size per query sent through the network, query processing time, change notification time, memory usage, CPU usage*}
- **Average** {*message size per query sent through the network, query processing time, change notification time, memory usage, CPU usage*}

The maximum query count for a processed query type with sub-selectivity was calculated after all iterations of a test scenario were completed. The following components were added to the system to store and serve the acquired data:

- A Redis instance deployed as a server for storing and managing measured actor and query counts. Because each node has its own JVM and actors process concurrently on each node, calculating these metrics without a mediator component would be hard.
- A middleware actor component that stored query and actor counts into the Redis store.
- A middleware actor component named as MetricsListener that retrieved CPU and memory usage using sigar-loader utility of kamon-io library (https://kamon.io) which is a comprehensive monitoring library that works fully integrated with Akka. We use ClusterMetricsExtension which is also suggested in Akka documentation (https://doc.akka.io/docs/akka/current/cluster-metrics.html) for efficently monitoring CPU and heap usage of each cluster node.
- Logger components of the actor system.
- Size estimator component of Apache SPARK library (https://spark.apache.org) for calculating SPARQL query and results size in order to reveal the network cost. Apache SPARK is built for large scale data processing in which the efficient memory management takes crucial role. Therefore it has built in and well designed components such as SizeEstimator used to estimate the size of both objects and datasets. By this way we can accurately evaluate the network costs in order to transmit data through network more efficiently.

### Data analysis

The system's behavior and performance are examined in this section. Therefore, the structure of the queries and collected data will be presented first. Following that, the behavior of the system will be analyzed by interpreting the collected data. In "Answers to

Knowledge and Requirements Satisfaction Questions" knowledge questions which are used as a guide during the study will be answered. Finally, the findings will be generalized to similar implementations of the abstract architecture using analogic reasoning in "Results and Analogic Generalizations".

The appendices B, C and D (see Appendices given as Supplemental File) contains a list of all the data collected for the evaluation. For each query selectivity type, measurement metrics are divided into three categories and presented as tables. The first type of table contains query and performance monitoring results. The second type of table displays the cluster's system resource usage results. The final one displays the network cost resulting from the size of messages sent during communication between cluster nodes of the actor system. The middleware component MetricsListener monitored and logged CPU and memory usage ratios. The result message size was calculated using the Size Estimator component of the Apache SPARK library. Results in tables were calculated by analyzing log files for each iteration of the evaluation. The source code of the *log analyzer* which performs analyzing the log files can be accessed *via* the following repositories: https:// github.com/seagent/loganalyzer and https://doi.org/10.5281/zenodo.10679278. All log files of the evaluation results and all analysis result files calculated by the log analyzer are hosted under: *resources/monarch-results* folder in the log analyzer source code.

Query and monitoring performance metrics listed within Appendix-B (see Appendices given as Supplemental File), in Table B1, B2, B3, B4 and B5 are average & maximum query processing time and change notification time. The elapsed unit of time between registering a (federated) query and receiving the final result is referred to as query processing time. Change notification time, on the other hand, is about monitoring performance and represents the unit of time between detection and notification of a result change. Second, the system resource usage metrics listed within Appendix-C (see Appendices given as Supplemental File) and shown in Table C1, C2, C3, C4 and C5 represent average and maximum Memory and CPU usages in Gigabytes and CPU threads, respectively. Memory usage is the amount of heap memory used by MonARCh during runtime, and CPU usage is the number of CPU core threads used by MonARCh. Finally, network result cost metrics listed within Appendix-D (see Appendices given as Supplemental File) and depicted in Table D1, D2, D3, D4 and D5 are query count per minute & query count per second registered to the system, total message size per second, maximum & total message size per query and total message size sent through the network without HTTP compression. DaDy change frequency metadata of the local endpoints used in the conducted experiment is also defined in Table 3.

The first query type is the Least Selective Query given in Listing-A2. It is made up of two SERVICE clauses, the first of which queries DBpedia and NyTimes and the second of which queries Stockmarket. It searches stock information for companies of specific types. The total count for the first SERVICE clause is 10,000, with 5,000 from DBpedia and 5,000 from NyTimes. The result for the second SERVICE clause is 5,000 as well. The least selective query type is intended to cover multiple endpoints for the same SERVICE clause while allowing for maximum result selection. There is only one query template with 0% filtering for this selectivity because its query structure is not suitable for filtering SERVICE

**Table 3 Dataset endpoint change frequencies.**

| Dataset endpoint | Frequency | Explanation |
| --- | --- | --- |
| DBPedia | NoUpdate | Data never changes |
| NyTimes | MidFrequentUpdates | Data changes from one a week to a couple of months |
| Stock | HighFrequentUpdates | Data changes once a day or more frequent |

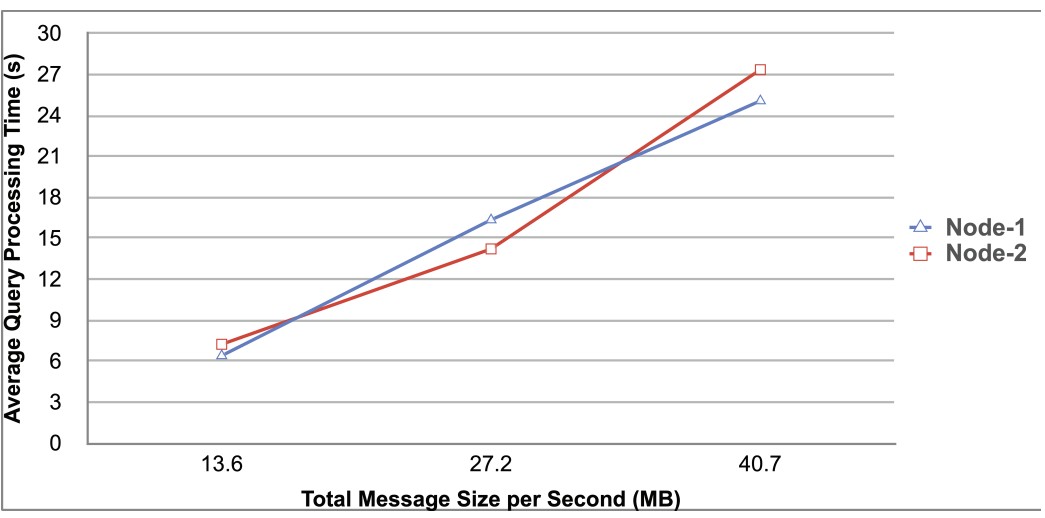

**Figure 4 Correlation between network bandwidth usage and query processing performance for the least selective query.**

clause results. While both result sets were quite large, the total message size per query sent over the network became large as well, as shown in Table D1. The relationship between average query processing time and total message size per second for the least selective query type is depicted in Fig. 4. The average query processing time increased rapidly as the total message size per second increased. Because the message size sent over the network per query was quite large (82 MB), using more network bandwidth and system resources caused the system to overload and slow down. As a result, the total number of monitored queries remained at 300.

Another query type shown in Listing-A3 is called Low Selective Query. It contains three SERVICE clauses, each of which is queried over DBpedia, NyTimes, and Stockmarket, respectively. Each SERVICE clause returns 5,000 results when no result filtering is used. This query searches for news and stock information for companies with predefined "reputation", "number of employees" and "market" property values, which can be used to filter the SERVICE clause results. A 20% percentage of the results returned by each SERVICE clause can be cropped at a time. Therefore five query templates can be generated, with each SERVICE clause to return starting from 5,000 to 1,000 respectively. As shown in Table D2, when no SERVICE clause result filtering was used, the total message size per query sent over the network increased to 80 MB, which was the same size as the least selective query template. When SERVICE clause result filtering was applied, the total
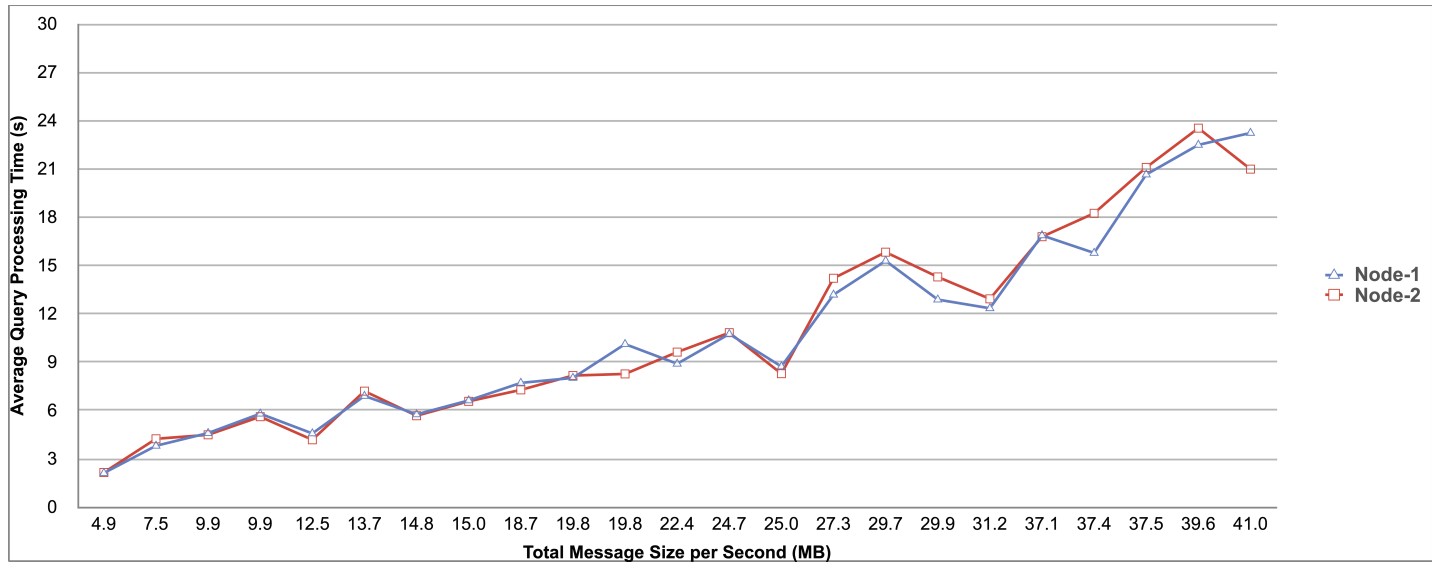

**Figure 5** Correlation between network bandwidth usage and query processing performance for the low selective query.

message size per query decreased, as shown in Table D2. On the contrary, in this way, the number of monitored queries increased up to 1,500 queries when SERVICE clause result filtering reached 80%. Figure 5 shows the relationship between average query processing time and total message size per second for the low selective query type. When the total message size per second increases average query processing time tends to increase rapidly.

The third query type is Middle Selective Query which is shown in Listing-A4. It also consists of three SERVICE clauses which are related to DBpedia, NyTimes and Stockmarket respectively. Middle Selective Query retrieves stock market data, as well as article and employee, counts for a specific company. Since the final generated result will have data about only one company, Middle Selective Query can actually be considered to be the most selective query. However, we named query selectivities based on the total SERVICE clause result selection and join cost of the hash join algorithm used in MonARCh's query execution engine. Thus, when no result filtering is applied, the first SERVICE clause returns 1, and each other SERVICE clause returns 5,000 results. Similarly to Low Selective Query, filtering the "reputation" values of the NyTimes related SERVICE clause and the "market" values of the Stockmarket related SERVICE clause can be used to gradually reduce the result count by 20% at a time. Table D3 shows that when no SERVICE clause result filtering was used, the total message size per query was cut in half to 40 MB when compared to the Low and Least Selective Query results; additionally, MonARCh could monitor up to 500 queries. When SERVICE clause result filtering was used, the total message size per query was slightly reduced to 7.7 MB, while the number of monitored queries was slightly increased to 2,500. Figure 6 shows the relationship between average query processing time and total message size per second for the middle selective query

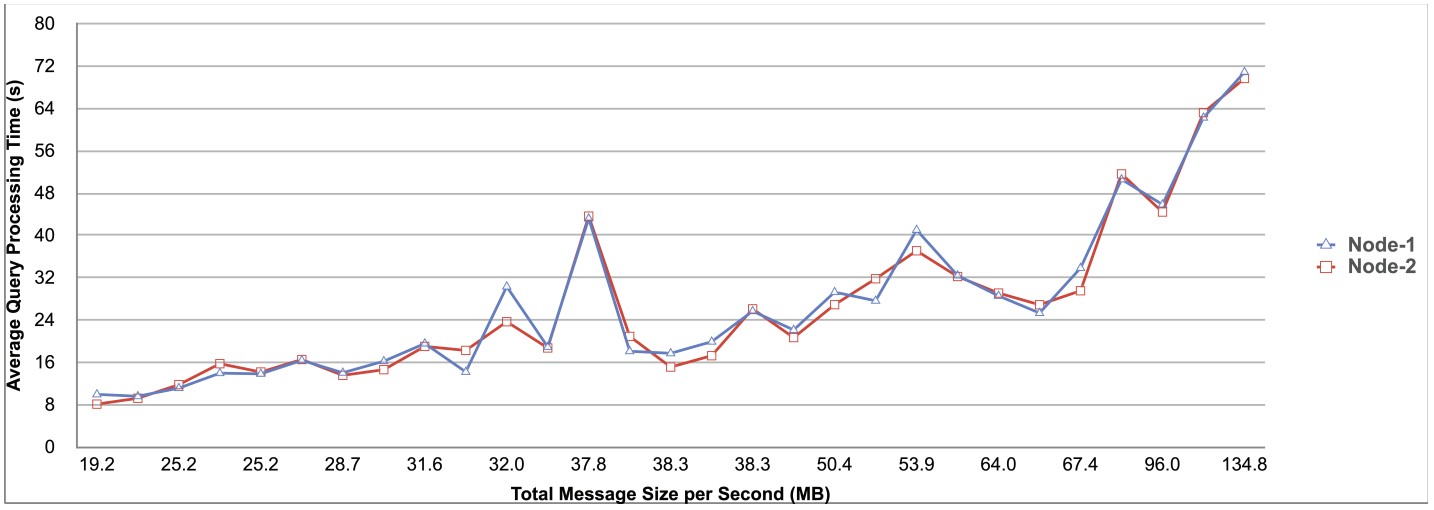

**Figure 6** Correlation between network bandwidth usage and query processing performance for the middle selective query.

type. An increase in total message size per second leads to a rise in average query processing time.

High Selective Query type shown in Listing-A5 contains three SERVICE clauses which are paired with one of the three datasets of our evaluation setup. It receives news, descriptive, and detailed stock market data for specific companies. When no result filtering is used, each SERVICE clause returns 500 results. While the SERVICE clauses for DBpedia and Stockmarket can be gradually filtered by 20% to 80%, the SERVICE clause for NyTimes can only be filtered by 80%. Unlike previous query types, the total result count reduced drastically even without SERVICE clause result filtering. As can be seen in Table D4 total message size per query was reduced to 1/5 according to the Middle Selective Query type without filtering and MonARCh could monitor 2,500 queries. Filtering SERVICE clause results by 80% reduced the total message size per query down to 1.7 MB and increased the number of monitored queries up to 12,500. Figure 7 shows the relationship between average query processing time and total message size per second for the high selective query type. As previously observed, as the total message size per second increases, so does the average query processing time.

Lastly, the Most Selective Query type depicted in Listing-A5 has also DBpedia, NyTimes and Stockmarket related SERVICE clauses. This query type returns stock market data as well as the article count value of a specific DBpedia company. As a result, the DBpedia related SERVICE clause returns a single result, just like the final result of this federated query. On the other hand, the remaining two SERVICE clauses return 500 results each if no result filtering is applied. While the Stockmarket relevant SERVICE clause can filter 20% to 80% of results, the NyTimes relevant SERVICE clause can only filter 80% of results. When compared to the High Selective Query type, network and communication costs were cut in half while the number of queries monitored was increased to 5,000. When SERVICE

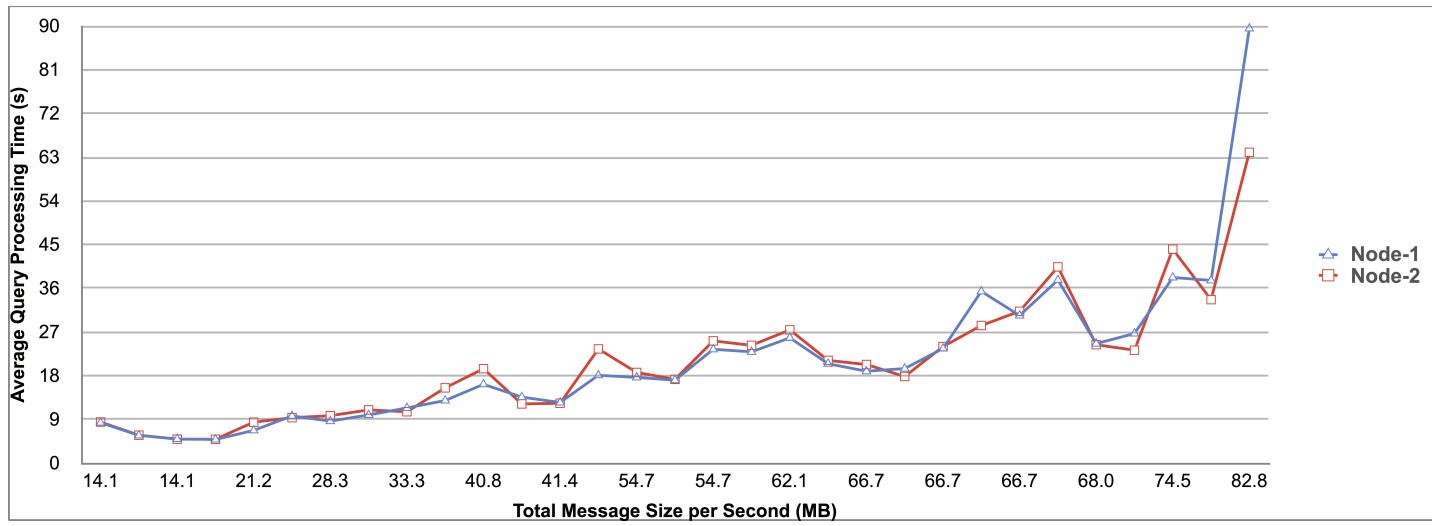

**Figure 7 Correlation between network bandwidth usage and query processing performance for the high selective query.**

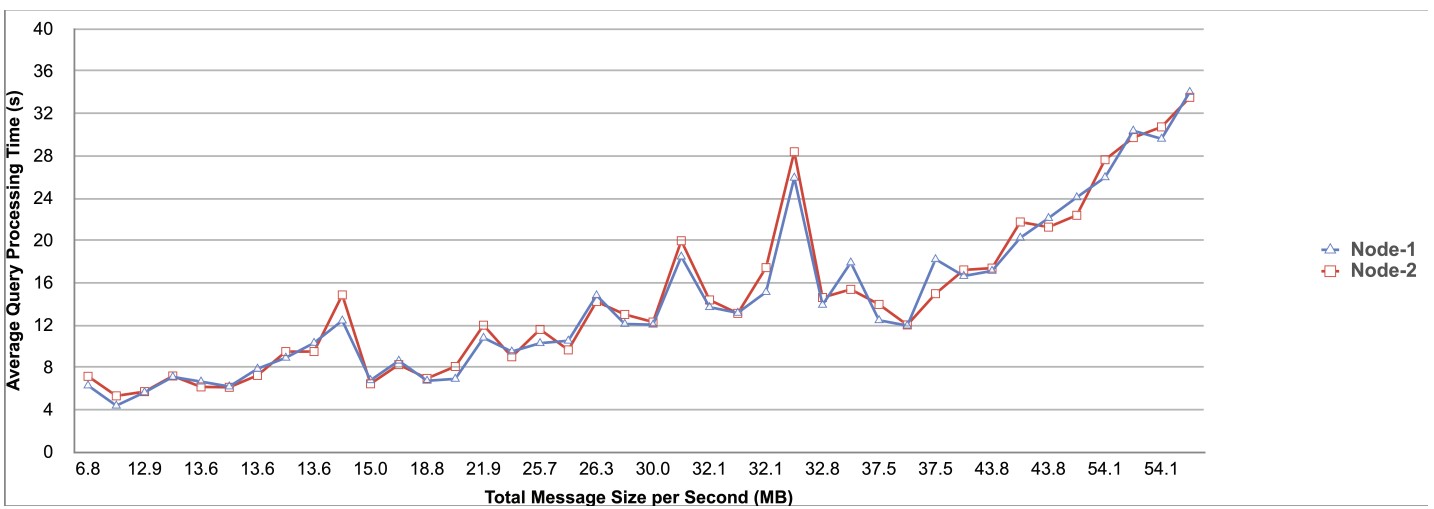

**Figure 8 Correlation between network bandwidth usage and query processing performance for the most selective query.**

clause results were gradually filtered down to 80%, the total message size per query was reduced to 0.81 MB, and the number of monitored queries was increased to 25,000. The relationship between average query processing time and total message size per second for the most selective query type is depicted in Fig. 8. Also, as the total message size per second increases, the average query processing time begins to lag rapidly.

As a result of the findings, we concluded that system resources and network metrics, particularly the amount of maximum heap memory and total message size sent over the network, limit the number of monitored queries. As shown in Fig. 9, as query selectivity

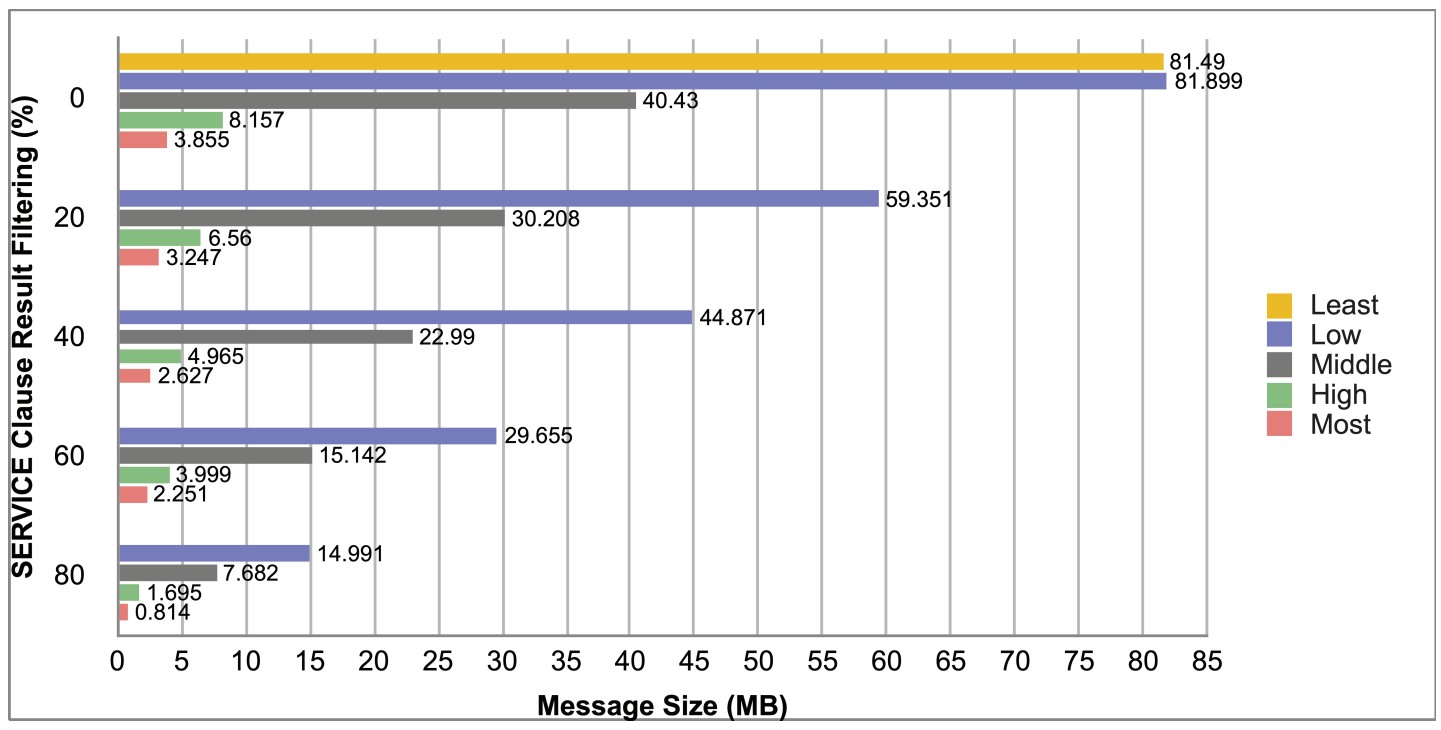

**Figure 9** Effect of increasing selectivity to the total msg size per query sent through the network according to the diverse type of query selectivities under the constant number of monitored queries as 500.

and SERVICE clause result filtering ratio increase, the total message size per query sent through the network decreases. Similarly, as shown in Fig. 10, as queries become more selective and their SERVICE clause results are reduced, the number of monitored queries increases. According to these graphs, the metrics "total message size per query" and "number of monitored queries" have a negative correlation. Furthermore, the maximum amount of assigned heap memory appears to limit the number of monitored queries when the system resource usage tables of Table C1, C2, C3, C4, C5 are examined. As used heap memory approaches the maximum amount of available memory, the actor system cluster becomes overloaded and eventually fails.

On the other hand, there is a correlation between the metrics of Average Query Processing Time and Total Message Size per Second when we analyze query and monitoring performance in Appendix Section B & network cost results tables in Appendix Section D. The metric "total message size per second" is calculated by multiplying the two metrics "query per second" and "total message size per query". It is also about the usage of network bandwidth. Therefore, as network bandwidth usage increases, endpoints must answer more queries at the same time, resulting in an increase in average query processing time. As the final note for this analysis; Figs. 4–8 show that correlation between "total message size per second" and "average query processing time" is straight and regardless of query selectivity.

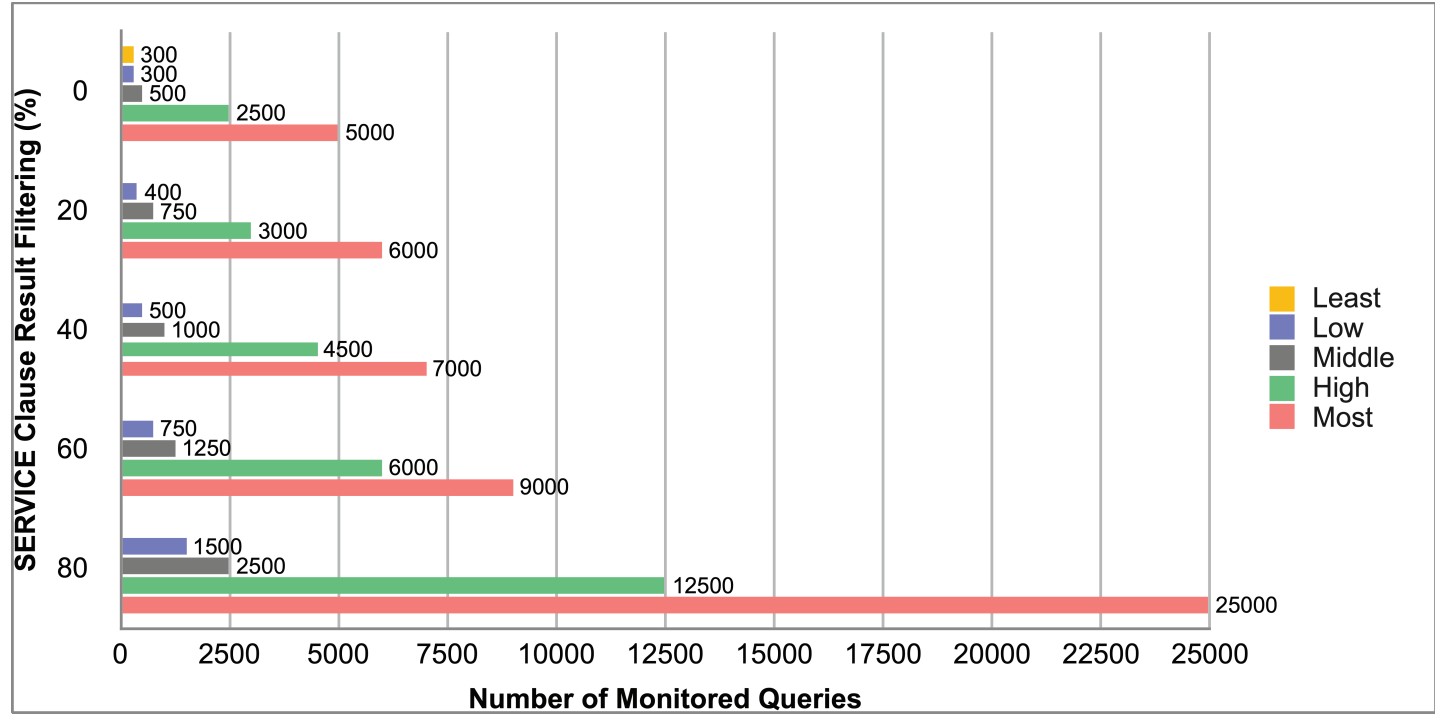

**Figure 10** Query monitoring capacity of MonARCh according to diverse type of query selectivities.

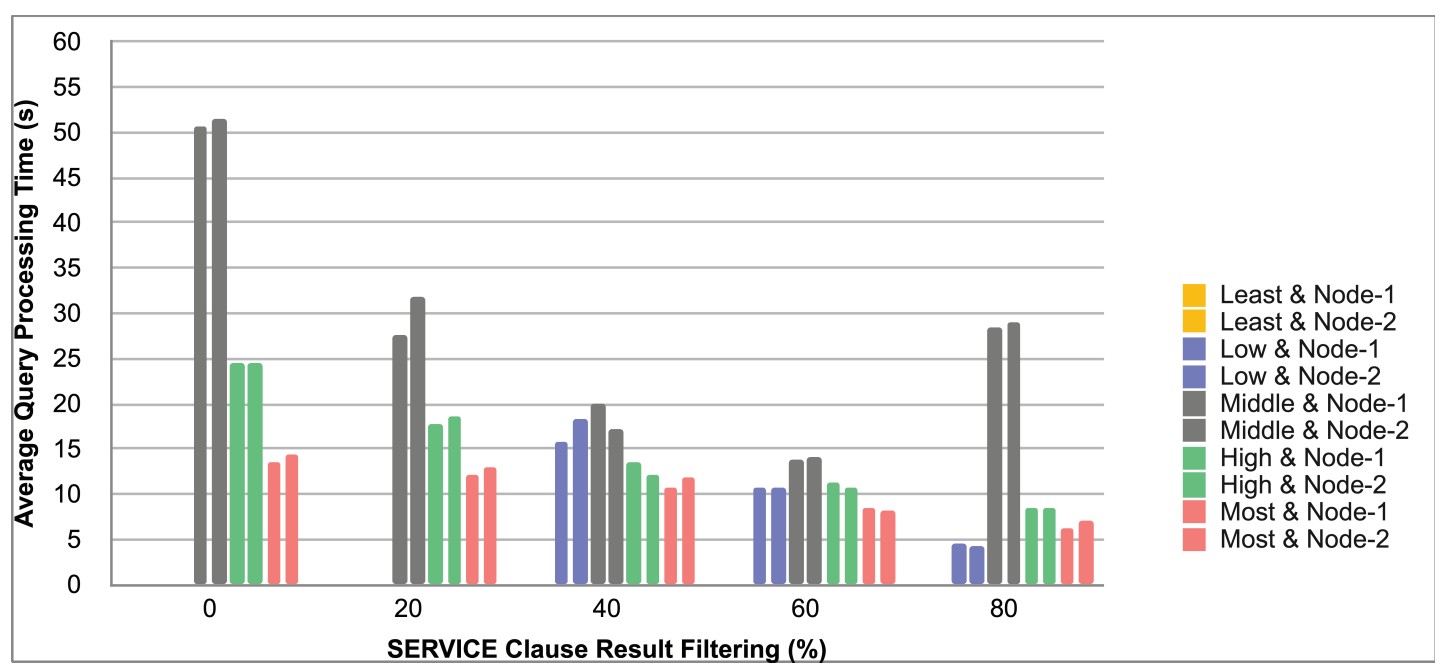

**Figure 11** Effect of increasing selectivity to average query processing time under the constant number of monitored queries as 500.

Finally, we compare the critical evaluation metric values as selectivity and SERVICE clause result filtering percentages increase while the number of monitored queries remains constant. We use 500 as the number of monitored queries because it is the most commonly used query count throughout the evaluation. Effect of selectivity and filtering is shown for the "average query processing times" in Fig. 11, for the "average change notification times" in 12, for the "average memory usage" in 13, for the "average CPU usage" in 14 and for the "total message size per query sent through the network" in 9. All of these graphs show that MonARCh performs better for more selective queries. Increasing selectivity by filtering more SERVICE clause results yields much better metric values, further improving the performance.

### Answers to knowledge and requirements satisfaction questions

Questions about knowledge and requirements satisfaction posed at the start of "Evaluation and Validation of the Architecture" can now be answered in light of the data gathered during the experiments. In this section answers to these questions are discussed. First, answers to performance analysis questions will be given.

### Answers to performance analysis questions

#### 1. Was the expected number of notifications gotten?

The agent system that registers SPARQL queries to the MonARCh was programmed to run according to the change frequencies of the New York Times and the Stockmarket datasets. It worked immediately following the updater programs in the evaluation. According to the findings, all newly registered or updated query results were notified to clients on time, with no result updates missing.

#### 2. What is the highest observed number of federated queries?

In Fig. 10 monitoring capacity of MonARCh is depicted according to diverse query types of base selectivity levels which increase progressively by filtering SERVICE clause results. Since there is no SERVICE clause result filtering with the least selective query system, it can monitor up to 300 queries. On the other hand under the maximum SERVICE clause result filtering ratio of 80% percent, by sending low, middle, high and most selective queries to the MonARCh system is able to monitor a maximum of 1,500, 2,500, 12,500 and 25,000 queries respectively.

#### 3. How many queries could be successfully registered per unit of time?

Network Cost Result tables for each query selectivity type listed in Appendix Section D depict the "query count per minute" and "query count per second" metrics. According to these tables, the "Maximum Query Count per Second" metrics with the maximum possible selectivity using SERVICE clause result filtering are "0.5", "2.5", "12.5", "16.666", "16.666" for queries of all selectivity types.

#### 4. What limits the number of queries being monitored?

The number of federated queries MonARCh can monitor is primarily determined by two factors: query selectivity and heap memory availability. These two variables are inextricably linked. While query selectivity affects the message size per query sent through the network, lower values for this metric yield more queries being monitored by MonARCh. On the other hand, increasing the amount of memory available to the system

either by adding more physical memory or by adding new nodes to the cluster makes the system monitor more queries.

**5. *What limits the number of queries being registered per unit of time?***

Because the actor system has a network bandwidth, there is a limit to the number of results that can be sent through the network at the same time, as indicated by the "total message size per second" value. Because this metric is related to "total query count per second" and "total message size per query" it is calculated by multiplying them. Reduced message size per query through more selective queries increases the total query count per second. As a result, query selectivity and network bandwidth both constrain the number of queries registered to the system per unit of time.

**6. *How does the number of SERVICE clauses in a federated query affect the performance of the system?***

Federated queries are composed of SERVICE clauses, each of which is linked to one or more SPARQL endpoints. SERVICE clauses mostly share query variables to connect with one another. A join algorithm is used to join the results of two queries that share a common query variable. The main result is generated after all SERVICE clause results have been joined. If a federated query contains more SERVICE clauses, more join operations per federated query are required. To take advantage of concurrency, MonARCh employs a parallel hash join algorithm to join SERVICE clause results at the same time, resulting in the quick generation of the main result. Even when a parallel join algorithm is used, an increase in the result count of the SERVICE clause in a federated query requires more CPU computation and more memory to be occupied. Therefore, adding new SERVICE clauses raises the total message count per query, limiting the number of SERVICE clauses that can be monitored.

**7. *What happens if the query selectivities change?***

The selectivity of triple patterns in a query affects the size of the results to be retrieved. A more selective SERVICE clause in a federated query produces a smaller result in size than a less selective one, as shown in Fig. 9. Reducing the result size of a query, on the other hand, increases MonARCh's monitoring potential, as shown in Fig. 10. Finally, increasing selectivity while keeping the number of monitored queries constant improves system metrics such as average query processing time (see Fig. 11), average change notification time (see Fig. 12), average memory usage (see Fig. 13), average CPU usage (see Fig. 14) and total message size per query (see Fig. 9).

**8. *What happens if query registration requests get more frequent?***

As discussed before "query count per second" directly affects the "total message size per second". Thus registering more queries per unit of time increases the total message size per second resulting in more network bandwidth usage. Sending more queries per second to the system causes endpoints to handle more queries at once. This reduces system performance in areas like query processing and change notification times. In Figs. 4–8 it is clearly seen that negative effect of increasing usage of bandwidth to the query processing performance.

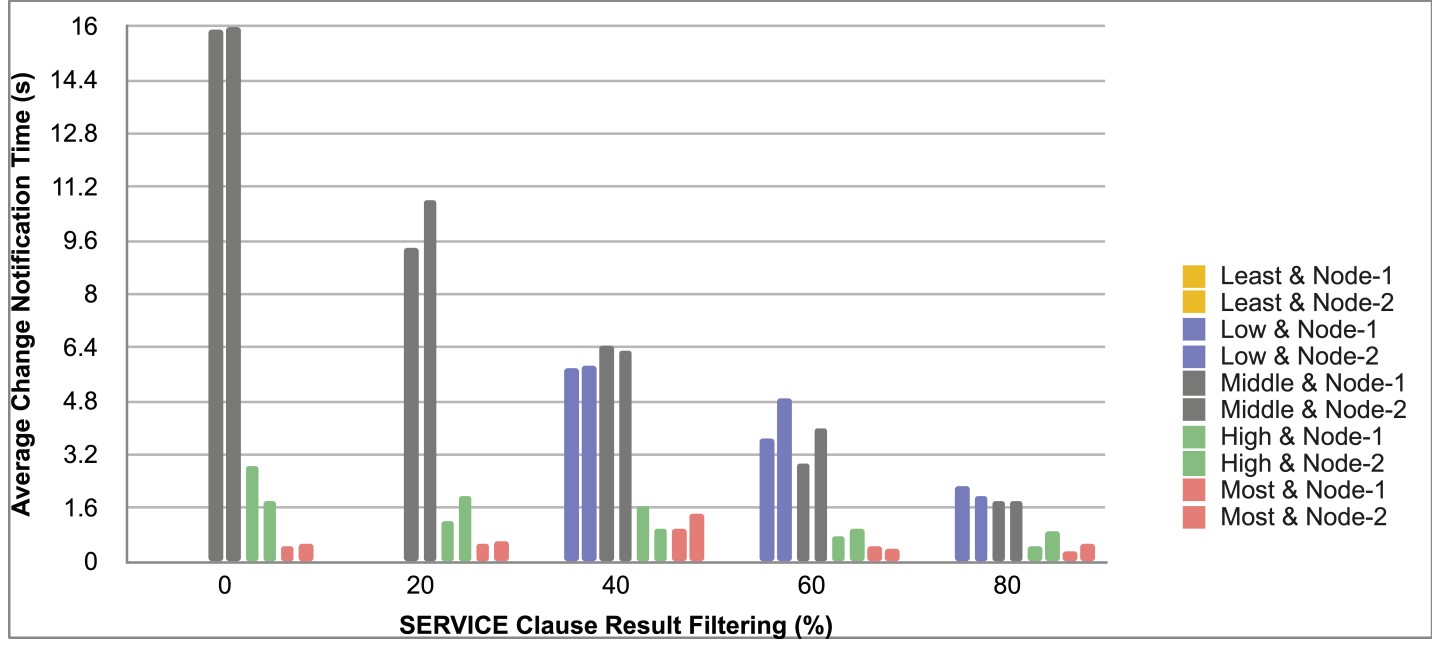

**Figure 12** Effect of increasing selectivity to average change notification time under the constant number of monitored queries as 500.

### 9. What are the network limits of the system?

When all network cost tables in Appendix Section-D are examined, the total message size sent through the network appears to reach a maximum of 134.776 MB. This value is calculated by multiplying the query count as 3.333 and the total message size per query as 40.43 MB. It is depicted in Table D3 and also in Fig. 6 generated for the middle selective query type.

### 10. Which factors affect the usage of the system resources?

When we examine the Fig. 9 total message size per query sent through the network is affected by the query selectivity. Thus more selective queries yield less usage of the memory, consequently enabling system to monitor more queries as can be seen in Fig. 10. Also Figs. 13 and 14 show the effects of query selectivity (while keeping number of monitored queries constant as 500) to the average memory usage and average CPU usage respectively. We can clearly devise from the graphics that increasing the selectivity of the query results better usage of the system resources.

On the other hand, the system consumes memory and CPU resources while keeping track of the changes by storing old version of the result sets. CPU consumption is relevant to the number of CPU bound actors working in the system. On the other hand the message size per query is also the total size of the result sets per query allocated in the memory. It is calculated by summing service clause results with the final federated query result. This is the actual memory consumption factor of our system which limits the number of queries being monitored. Besides while performing hash join operation, the system consume additional memory and CPU either. But this is only temporarily because these actors are

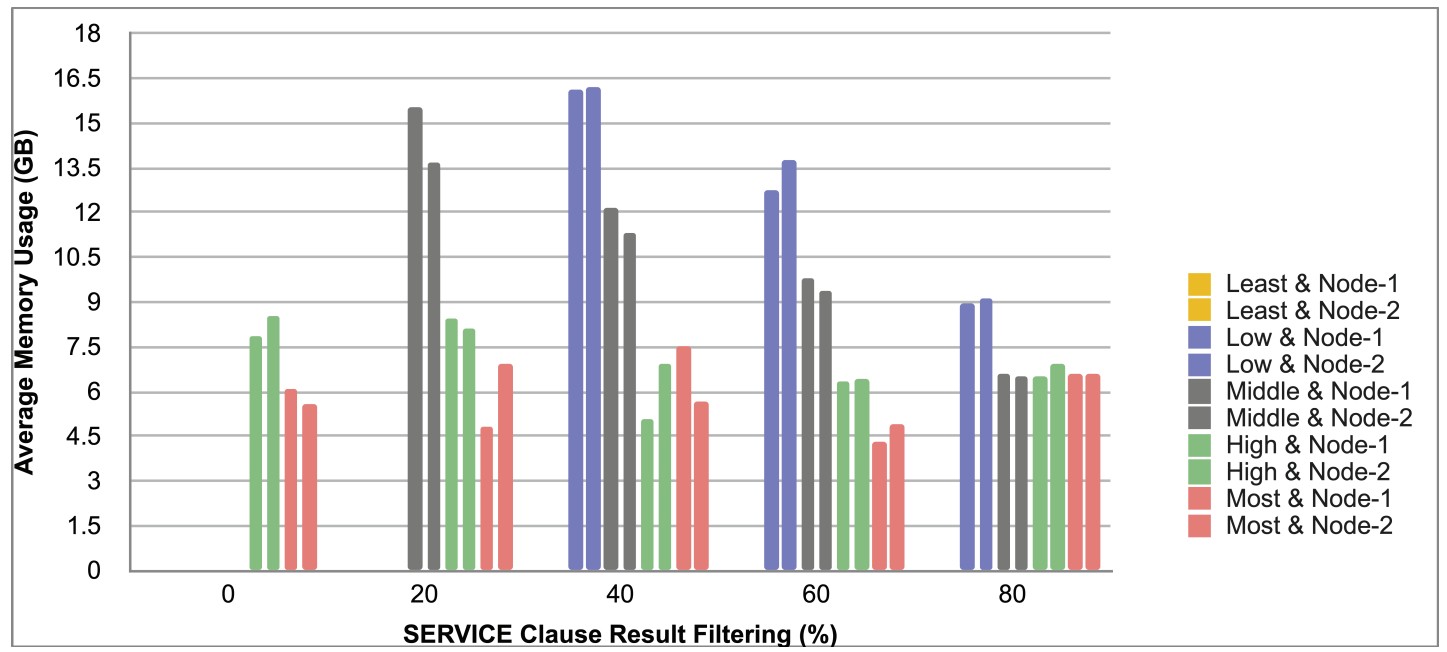

**Figure 13** Effect of increasing selectivity to average memory usage under the constant number of monitored queries as 500.

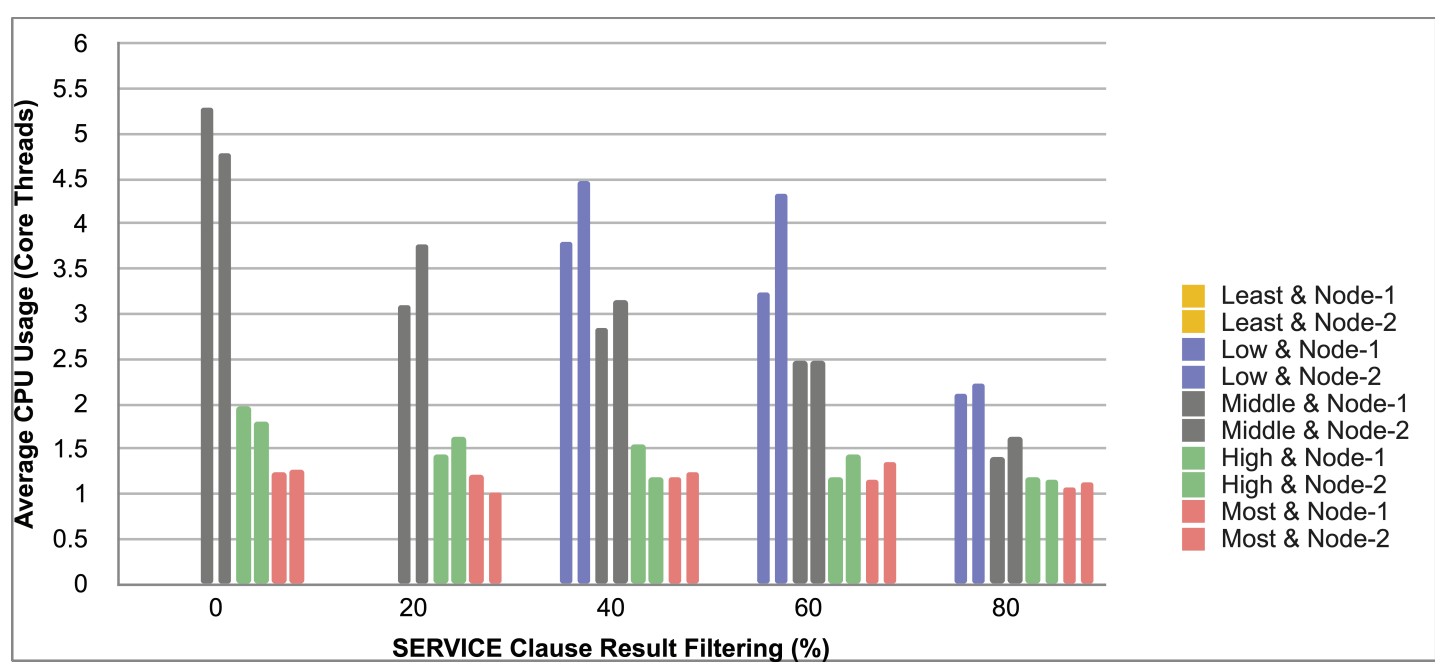

**Figure 14** Effect of increasing selectivity to average CPU usage under the constant number of monitored queries as 500.

immediately killed after the final result has been generated. Also hash join actors are not created using cluster sharding like query responsible actors. Because hash join operation has already been parallelized through the Federator actor, also there is no need to distribute hash join actors of a particular SPARQL query anymore. Thus, the hash join operation relevant with a query is performed on the same node with the responsible Federator actor. This prevents the system from a potential network bottleneck.

Secondly, answers to design discussions questions will be given below.

*Answers to design discussion questions*
*1. How would the artifact be affected if a different join algorithm was used?*

A hash join algorithm called grace join was used in the implemented system to produce the main result by using intermediate results. Grace hash join divides intermediate results into much smaller buckets based on a mathematical function to generate a hash map and uses concurrency to join buckets with the same key at the same time. In many cases, dividing a large join operation into smaller jobs and utilizing all cores and processing power of the CPU reduces the join time. This method works well with the actor-based system architecture. On the other hand, when intermediate results are too big to fit into the memory and skewed, hash join performance starts to drop because of the join operation cost. Bind join makes much more sense in this scenario because it binds one intermediate result of a SERVICE clause into another before querying to narrow the intermediate result space.

*2. How would the artifact be affected if the actor model was not used?*

There are three main reasons for choosing the actor model for concurrency:

I. The actor model imposes immutability. When multi-threaded applications become mutable, many bugs and errors occur at runtime, necessitating the use of a locking mechanism.

II. The designed system requires actor systems to be easily distributed and scalable.

III. In concurrent systems, synchronization is a major issue. Because actors communicate with one another by sending messages, they fully encapsulate their behavior and state. This avoids the synchronization and resource allocation issues that are common in concurrent systems.

Linked Data query systems need high parallelism, that's why systems like Comunica (*Taelman et al., 2018*) are implemented using the actor model. Thus, the advantages of actor systems over other concurrency models make them the superior choice for the implemented Linked Data artifact. If another concurrency model had been chosen, it would be hard to avoid synchronization problems, resource allocation errors and scaling issues.

There are several actor model languages and frameworks that can be used to implement an actor model. The key point is that the actor framework should be mature, robust, well documented, simple to use, and resource (computer) friendly. For example, the Erlang and Elixir programming languages rely on the Erlang virtual machine, whose threads are much lighter than Java threads. But on the other hand, Erlang and Elixir are both fully functional

programming languages which most developers are not familiar with, therefore this may affect the development process in a bad way. When the implementation difficulty has been put aside, from the system performance point of view Elixir or Erlang can be seen as a replacement for the Akka toolkit. On the other hand, there are emerging alternative actor frameworks such as Riker (https://riker.rs) written in Rust programming language, CAF (http://actor-framework.org) written in C++, Comedy (https://github.com/untu/comedy) built for Node.js, Orleans (https://dotnet.github.io/orleans) built for .net framework. Riker has solid concurrency and memory management thanks to Rust's thread safe unique ownership model and built in memory safety guarantee which makes it a good alternative for developers familiar with Rust. On the other hand, CAF has shared nothing architecture to completely prevent race conditions and lightweight actors thanks to efficient memory management of C++. However, when compared to Akka, Erlang, and Elixir, they all have a less mature multi-thread mechanism or a more limited development community. Orleans brings full power of .NET platform along with thread pool support and unique abstractions such as Grains (virtual actors) and Silos (group of grains) enhancing the scalability and building a cluster which makes it good candidate for replacement to Akka for C# developers. Finally, Comedy provides a good alternative for Node.js and Javascript developers providing good concurrency support and scalability with cluster management under its newly growing ecosystem.

**3. What happens if a query should be answered by a SPARQL endpoint with no DaDy definition?**

VoID descriptions about SPARQL endpoints are used to store DaDy change frequencies. If a dataset has no DaDy description then a default change frequency can be set to maintain the monitoring job.

**4. What happens if a query is answered by a SPARQL endpoint that lacks a VoID definition?**

Some federated query engines like SPLENDID (*Görlitz & Staab, 2011*) and WoDQA use VoID descriptions for constructing federated SPARQL queries. In this study VoID descriptions of datasets are used because WoDQA uses this metadata for fast query analysis and federated query generation. Since SPARQL endpoints do not need to have a VoID description. Thus an alternative query federation engine which is not based on VoID descriptions like FedX (*Schwarte et al., 2011*), ANAPSID (*Acosta et al., 2011*), FEDRA (*Montoya et al., 2015*) or HiBISCUS (*Saleem & Ngomo, 2014*) can be used instead of WoDQA. However, this may affect the federated query's generation time. Other works on SPARQL query federation are not mentioned here because they do not have downloadable source code or binary releases that can be used by a newly developed system.

After answering knowledge questions, requirements satisfaction questions can be answered.

*Answers to requirements satisfaction questions*
**1. Does the artifact isolate developers from the details of Linked Data monitoring? (What assumptions does the artifact make about the Linked Data monitoring effort required by the developer?)**

Empirical results show that by creating agent applications that issue and register SPARQL queries to MonARCh, developers can easily monitor the desired subset of Linked Data. In this way the monitoring process is handled by MonARCh, also the developers are not required to know about details such as detection and notification of changes. However, developers should be familiar with the Linked Data environment context, SPARQL syntax, the actor model, and the development of actor-based applications that are compatible with Akka.

***2. How long does it take between detecting a query result change and notifying the registered applications? Does it meet all of the functional requirements?***

Tables B1, B2, B3, B4 and B5 show the average change notification times based on query selectivity levels. As previously discussed and analyzed, change notification times increase, as does average query processing time, when more queries are sent to the system in a given unit of time. When these metric values are compared to the change notification times listed in query processing and monitoring performance tables in Appendix Section B, a direct correlation can be established. MonARCh would have reasonable change notification times if the query count per second metric was adjusted correctly and the total message size per second metric was kept under control based on the value of the total message size per query.

***3. How much time elapses between a query result change and the detection of a change? Does it meet all of the functional requirements?***

Two dataset updater programs have been set to update *2f* per hour for stockmarket and *f* per hour for NyTimes for the use case scenario. The client actor program, which registers query monitoring requests to the cluster, has been configured to run immediately after the updater programs have completed the first update, allowing MonARCh to be synchronous and detect changes immediately without missing any updates.

### Results and analogic generalizations

A validation model can be generalized to the population it represents if it is able to support valid analogic inference. To determine whether the proposed analogic inference is valid, researchers should ensure that the prototype system is sufficiently similar to the population it represents (*Wieringa, 2014*).

To properly generalize to real-world scenarios, the system should be examined from three perspectives. To begin, in order to keep up with the high change frequency of datasets, the system is subjected to a heavy bursty query load in the use case scenarios described in "Evaluation and Validation of the Architecture". The expected change frequency of real-world datasets, on the other hand, is much lower. According to the DaDy specification (*Hausenblas, 2010*), even the dataset with the highest change frequency is expected to update a few times per day. As a result, the system is expected to be under a moderate query monitoring load. This allows the network driver of the actor system (Artery) to be requested by messaging actors at a much lower rate and eliminates waiting time caused by spending a large portion of network bandwidth. This also allows the cluster to maintain its functionality. According to the evaluation story, the message size associated with the query result change grows larger as the associated article count and stock price of

each company in the NyTimes and Stockmarket datasets are updated. Only a portion of the triple patterns in datasets are likely to be updated in real-world scenarios and in most cases. Consequently, as the size of the generated results decreases, so will the size of the overall message sent for each query through the network. Therefore, as shown in Fig. 10, this enables the system to monitor much more queries.

From the second point of view, it is easy to imagine a real-world implementation of the monitoring artifact having many different queries with different scenarios. These queries will also be of varying complexity, necessitating the use of a diverse set of SPARQL endpoints. We explained our evaluation story in the financial domain and our 21 test scenarios generated with the relevant query types of a diverse set of selectivities in "Evaluation Story and Test Scenarios". According to the findings, we first concluded that monitoring capacity is related to both system heap memory capacity and query & SERVICE clause selectivity. The total message count per second sent through the network, on the other hand, affects query processing and monitoring performance of the system, which increases or decreases with query count per second & total message size per query metrics. Therefore in practice, having queries with different types of selectivity levels and changing query loads can cause system performance to increase or decrease with a diverse set of queries.

Thirdly, the configurations of the computers used for the system implementation are not suitable for real life scenarios. In a real life case, much more powerful machines or a cloud is expected to be used which will enhance the performance. However, if we consider that our actor cluster is at the application level, building it on top of a cluster at the operating system level using Kubernetes (https://kubernetes.io) with Docker (https://www.docker.com) containerization and outfitting the physical machines with more network bandwidth would significantly improve system performance and advance it to the next level.

Given these three aspects of real-world cases, the data collected during experiments, as well as the answers to knowledge and requirement satisfaction questions devised using this data, can be used to make valid generalisations about the given abstract architecture. According to the evaluation results, MonARCh can easily operate in real-world scenarios, even with the experimental settings described in "Experimental Settings". Furthermore, if the system's heap memory is expanded and the size of messages sent over the network is reduced, system performance will obviously improve significantly.

### Comparison to the relevant systems

Systems in the monitoring context from the linked data point of view and from the pusblish/subscribe point of view are analyzed and elaborated in the Related Work section through the Tables 1 and 2 respectively. When we compare MonARCh to the systems listed in Table 1, it can clearly be seen that none of the systems other than MonARCh meets all of the requirements for monitoring in the linked data context. While DSNotify, SEPA and SPS monitor single SPARQL queries they are not capable of monitoring federated queries. Also because of configured to be work in local only they all have access to the management of the dataset related to their monitored queries letting them the exact

awareness of updates but are unable to work on the Web of Data. Moreover among these systems only SPS and MonARCh are scalable.

On the other hand when we compared to the systems listed in Table 2, MonARCh has many of features listed in this context making it one of the competitive state of the art systems among others. Firstly, all systems listed in this table are distributed and scalable. They all have fault tolerance other than STREAMHUB and SPS. From the processing point of view; only STREAMHUB, E-STREAMHUB, SPS and MonARCh are able to work parallel, concurrent and distributed which is important for completing their work faster. Among all these systems BlueDove, SEMAS and E-STREAMHUB have elasticity feature because they are configured to work on the cloud environment. But thanks to the robustness of the underlying architecture, both Kafka and MonARCh can easliy be configured and extended to become elastic. All these systems are aware of updates by acquiring the data directly from publisher thanks to their publish/subscribe architecture. However none of them are able to monitor federated SPARQL queries and work on the Web of Data. Accordingly, in addition to being one of the state of the art systems MonARCh is the only architecture which closes this gap in the linked data monitoring context. After all, LDN and WebSub are important extensions to Linked Data for sharing information between the data sources and consumers. As a bridge, widespread use of these protocols by the SPARQL endpoints and other linked data sources will enable the construction of the publish/subscribe systems over the Web of Data. Thus, extending the architecture of MonARCh in accordance to one of these protocols will be also a significant improvement leading to the open future directions.

## CONCLUSION AND FUTURE WORK

MonARCh, a scalable monitoring architecture for SPARQL queries in the Linked Data environment, is presented in this article. Users can subscribe SPARQL queries to MonARCh and are notified when the results of the subscribed queries change. It is designed as a push/pull approach that proactively executes SERVICE clauses over the endpoints based on DaDy change frequencies (pull), constructs the new result, and notifies related clients when a change is detected (push). We have implemented the system by following the design science methodology and using the Akka toolkit, an actor model implementation with the cluster sharding feature.

MonARCh's main contribution is the ability to monitor the results of the federated SPARQL queries over the dynamically changing datasets of the Web of Data. The proposed system works as parallel, concurrent and distributed on the cluster mode thanks to the actor model and the implemented parallel join algorithm. Moreover, our system is scalable and fault tolerant thanks to the cluster sharding feature of Akka. As demonstrated during the evaluation process, it can function as a cluster by allowing more computers to join in order to increase total system memory and computation power. This allows for monitoring more queries and scaling as needed.

While MonARCh is not elastic yet but the infrastructure of Akka naturally supports the elasticity, we are planning to add this feature later as future work. Moreover, we plan to add the Akka persistence feature to the MonARCh as future work, allowing the actor system to

self-heal when a node or cluster is shut down. This will complete the fault tolerance of the system in all aspects. And another future project is to create an efficient cluster sharding algorithm. This algorithm can balance the system load on nodes, improving system performance, scalability, and robustness.

The requirements, purpose and structure of MonARCh are very similar to the aforementioned pub/sub systems. As can be seen from Table 2 it differs from the state of the art pub/sub systems in the data acquisition mechanism from the data sources, for detecting the aforementioned changes. And also MonARCh's function is to isolate developers from the details of this process. In short, the system proposed in this study is, to our knowledge, the first one to serve developers in this sense. Consequently, there is no such an architecture covering all the requirements listed in "Requirements of the Linked Data Monitoring Architecture and Knowledge Questions" while providing most of the qualifications in Table 2 and being listed as one of the state of the art system.

From the limitations point of view we have faced several challenges both during the development and testing of the architecture. For example before cluster sharding we were using pure remoting feature of Akka in order to handle scalability manually. We scale to new node when a limit of actor size has reached to an extent and store full actor references in Redis server related to each federated query. Overwhelming use of Redis by many actors concurrently caused to a bottleneck in network traffic. Cluster sharding feature has become lifesaver for us to rebuild the entire architecture from scratch, slightly with less effort and efficiently. On the other hand that cluster sharding and actor model require high level of expertise to be able to configure and implement the architecture efficiently can be seen as a drawback. Each service clause result over the specific endpoint is stored and cached in the relevant executor which causes consuming the available memory faster can be seen as the main limitation of MonARCh. It can be serialized and kept within Redis and can be unleashed to the memory only during the join operation. But this can also burden a significant cost to Redis store and will be investigated in the upcoming research. During the early testing of MonARCh, serialization of the messages was a big problem which resolved by implementing specific serializer managing each type of message sent through the network. One last limitation of our research is building the cluster only upon two nodes which results to limited scalability especially on evaluation. Unfortunately our funding has let us to buy only two server nodes, but we are planning to expand the cluster size by adding new servers which possibly be funded by our new future research projects. Therefore we can re-evaluate MonARCh, observe the metrics and interpret the new results within the expanded cluster setup.

According to Klímek, Škoda & Nečaskỳ (2019)'s detailed Linked Data consumption survey, users have difficulty learning Linked Data technologies. Though MonARCh greatly assists developers with query monitoring, it still requires them to write SPARQL queries. Writing queries may be difficult for the developers unfamiliar with the SPARQL language or Linked Data tools. There are some machine learning-based studies for this problem, such as Soru et al. (2018), Steinmetz, Arning & Sattler (2019), Pradel, Haemmerlé & Hernandez (2013), Ochieng (2020). However, none of them are integrated with a tool like

MonARCh. As a result, another area of investigation could be the automated translation of natural language queries to SPARQL.

RDF streams (https://www.w3.org/community/rsp/wiki/RDF_Stream_Model) is another research direction for RDF-related research. Since the Web of Things gained popularity, the means for exchanging the streaming sensor data present on it in the form of RDF have also become an active research area. RSP (RDF Stream Processing) engines are systems that can perform RDF stream queries. CQELS 2.0 (*Le-Tuan et al., 2022*), Katts (*Fischer, Scharrenbach & Bernstein, 2013*), Distributed-Etalis (*Ren, 2016*), and Strider (*Ren & Curé, 2017*) are a few scalable RSP engines that support monitoring. In the future, we intend to incorporate a scalable RSP engine into our system. On the Sensor Web, there are ontological recommendations for defining metadata and data about sensors, such as the Semantic Sensor Network ontology. Streaming sensor data can be monitored and combined with regular RDF data using this ontology and a scalable RSP engine. Later on, we plan to evaluate this new feature of MonARCh on a benchmark for RDF streams like SRBench (*Zhang et al., 2012*).

One other research area related to the Web of Things is simplifying IoT application and service development. SymBIoTe (*Jacoby et al., 2017*) is the result of such an effort. Its main purpose is to offer interoperability and mediation services on present IoT platforms. A SymBIoTe enabler is one of the many components forming a SymBIoTe system (*Kusek et al., 2017*). Its main purpose is fetching domain specific data from SymBIoTe enabled IoT platforms and providing real time data processing services. One future work for MonARCh is providing its services in a SymBIoTe enabler wrap, where regular RDF data can be combined with sensor data coming from SymBIoTe compatible platforms.

One final future research direction is to define the change frequencies of data sources that do not have DaDy definitions. Many SPARQL endpoints do not have DaDy definitions. However, there are some studies (*Dividino et al., 2014*) aim to analyze and define the dynamics of Linked Data sets. These studies could be combined with MonARCh to define DaDy metadata for the data sources that lack change dynamics information, allowing MonARCh to monitor them as well.

### Funding
The authors received no funding for this article.

### Competing Interests
The authors declare that they have no competing interests.

### Author Contributions
- Burak Yönyül conceived and designed the experiments, performed the experiments, analyzed the data, performed the computation work, prepared figures and/or tables, authored or reviewed drafts of the article, and approved the final draft.

- Oylum Alatlı conceived and designed the experiments, authored or reviewed drafts of the article, and approved the final draft.
- Rıza Cenk Erdur conceived and designed the experiments, authored or reviewed drafts of the article, and approved the final draft.

## Data Availability

The source code of MonARCh is available on GitHub and Zenodo:

- https://github.com/seagent/monarch

- Yönyül, B. (2024). MonARCh: Monitoring Architecture for SPARQL Result Changes (https://github.com/seagent/monarch.git) Zenodo. https://doi.org/10.5281/zenodo.10679208.

The source code of the dataset generator and updater which generates and updates the evaluation data used by MonARCh is available on GitHub and Zenodo:

- https://github.com/seagent/datasetupdater

- Yönyül, B. (2024). DatasetUpdater: An Artificial RDF Dataset Generator And Updater (https://github.com/seagent/datasetupdater.git) Zenodo. https://doi.org/10.5281/zenodo.10679264

The source code of the log analyzer which also contains the evaluation log results is available on GitHub and Zenodo:

- https://github.com/seagent/loganalyzer

-Yönyül, B. (2024). LogAnalyzer: Log Analyzer for MonARCh (https://github.com/seagent/loganalyzer.git) Zenodo. https://doi.org/10.5281/zenodo.10679278.

## Supplemental Information

Supplemental information for this article can be found online at http://dx.doi.org/10.7717/peerj-cs.2133#supplemental-information.

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
