# Peer review of "MonARCh: an actor based architecture for dynamic linked data monitoring"

_PeerJ Computer Science, doi:10.7717/peerj-cs.2133_

## Round 0.1 · original submission · Minor Revisions

Your paper can be accepted if you make the revisions recommended by the reviewers.

Reviewer 1 ·

Basic reporting

This paper presents a novel design MonARCh, an architecture designed to monitor changes in SPARQL query results within the Linked Data environment. Especially, it provides federated queries for the SPARQL queries.

Instead of the traditional push or pull-based, data platform, the MonARCh, uses a combined push and pull-approach to manage the data updates for the consumers

It provides sufficient literature reviews. The background, however, is lacking in this paper. The authors provide some basic information in the introduction section. It would be great if there were a background section to discuss basic information like SPARQL and federated queries. The English of this paper is clear and professional.

Experimental design

This paper provides a solid discussion of the design principles of the federated SPARQL data platform. It then proposed the MonARCH that meets these requirements. To achieve the federated queries, this paper proposed an architecture that has a so-called Parallel Join Manager to manage the hash join and Hash Join Performer to execute the join for distributed clusters.

It is a little bit unclear how exactly the Parallel Join Manager, the key of the entire architecture, works. Although an algorithm is provided in discussing the join method carried by the manager, it would be great if authors could provide more information (or a figure) about this component.


For the experiments setup, it doesn't show the scalability of this architecture with only two nodes provided. The author needs to demonstrate how well it performs when the number of nodes increases. As with more nodes, more connections are required and the performance of the proposed Join Mangaer and Join Performer will be challenged.

At the beginning of section 6, a total of fourteen questions are raised to discuss the performance of the design. The authors need to categorize these questions and discuss them in a more organized manner.
For example, some questions (like 9,12, 13, 14) are for performance analysis and some are for design discussion (like 10, 11).

Section 6.2 provides too much technical details (listings of data format and queries). The authors should summarize them and give a brief description. These listings should be put in the appendix.

Validity of the findings

The paper presents a novel architecture for monitoring Linked Data. It provides thorough evaluation sections with real datasets. However, section 6 needs a major revision to present the data and demonstrate the functionality and performance of this architecture in a brief and organized way. Besides, the evaluation only shows the case with two nodes, which I believe is too simple to demonstrate this architecture.

It would be great if the author could summarize the performance metrics and provide more cases (different number of nodes).

Cite this review as

Reviewer 2 ·

Basic reporting

The paper is well-written in clear and professional English. The technical details are explained in a way that is accessible and understandable. The literature review is thorough, providing a solid foundation and context for the study by referencing relevant and up-to-date resources. This establishes a clear connection to existing knowledge and frameworks in the field.

However, while the results are well-presented, the paper could strengthen its formal rigor by providing clearer definitions of all terms and more detailed explanations. Specific comments are annotated in the attached PDF.

Experimental design

The paper presents original primary research that aligns well with the aims and scope of the journal, particularly focusing on advancements in the Linked Data environment and SPARQL query monitoring. The methods section of the paper is meticulously detailed, providing comprehensive descriptions of the system architecture, implementation, and evaluation procedures. This level of detail ensures that the study can be replicated by other researchers, which is crucial for validating the findings and furthering research in the field.

Validity of the findings

The paper's findings are valid based on the detailed descriptions of the methodologies used and the extensive data provided. The conclusions are justified by the results and are relevant to the stated aims of the research. The paper ensures that all data and methods are open and replicable, contributes significantly to its validity, and aligns with the journal's emphasis on rigorous and transparent research. By allowing others to replicate and validate the findings, the paper strengthens its contributions to the field of Linked Data and SPARQL query monitoring.

Annotated reviews are not available for download in order to protect the identity of reviewers who chose to remain anonymous.
Cite this review as

---

## Round 0.2 · accepted · Accept

The authors have addressed the reviewers comments.